
# Quantification of the role of stabilized Criegee intermediates in the formation of aerosols in limonene ozonolysis

Yiwei Gong[1] and Zhongming Chen[1]

[1]State Key Laboratory of Environmental Simulation and Pollution Control, College of Environmental Sciences and
Engineering, Peking University, Beijing 100871, China

*Correspondence to:* Z. M. Chen (zmchen@pku.edu.cn)

**Abstract.** Stabilized Criegee intermediates (SCIs) have the potential to oxidize trace species and to produce secondary organic aerosols (SOA), making them important factors in tropospheric chemistry. This study quantitatively investigates the performance of SCIs in SOA formation at different relative humidity (RH), and the first- and second-generation oxidations of endo- and exo-cyclic double bonds ozonated in limonene ozonolysis are studied separately. Through regulating SCIs scavengers, the yields and rate constants of SCIs in reaction system were derived, and the amounts of SCIs were calculated. The amount of SOA decreased by more than 20% under low-humidity conditions (10–50% RH), compared to that under dry conditions due to the reactions of SCIs with water, while the inhibitory effect of water on SOA formation was not observed under high-humidity conditions (60–90% RH). When using excessive SCIs scavengers to exclude SCIs reactions, it was found that the effect of water on SOA formation with the presence of SCIs was different from that without the presence of SCIs, suggesting that SCIs reactions were relevant to the non-monotonic impact of water. The fractions of SCIs contribution to SOA were similar between dry and high-humidity conditions, where the SCIs reactions accounted for ∼ 63% and ∼ 73% in SOA formation in the first- and second-generation oxidation, however, marked differences in SOA formation mechanisms were observed. SOA formation showed a positive correlation with the amount of SCIs, and the SOA formation potential of SCIs under high-humidity conditions was more significant than that under dry and low-humidity conditions. It was estimated that 20–30% of SCIs could convert into SOA under high-humidity conditions, while this value decreased nearly by half under dry and low-humidity conditions. The contributions of limonene-derived SCIs to SOA in the atmosphere were evaluated, and it was estimated that the contribution of SCIs to SOA was the lowest under low-humidity conditions. Under high-humidity conditions, the contribution of limonene-derived SCIs to SOA was $(8.21 \pm 0.15) \times 10^{-2}$ µg m$^{-3}$ h$^{-1}$ in forest, $(6.66 \pm 0.12) \times 10^{-2}$ µg m$^{-3}$ h$^{-1}$ in urban area, and $(3.95 \pm 0.72) \times 10^{-1}$ µg m$^{-3}$ h$^{-1}$ in indoor area. Water was an uncertainty on the role of SCIs playing in SOA formation, and the contribution of SCIs to SOA formation needed consideration even under high RH in the atmosphere.

## 1 Introduction

Stabilized Criegee intermediates (SCIs), formed from the stabilization of excited Criegee intermediates (ECIs) during the ozonolysis of alkenes, play important roles in atmospheric chemistry (Criegee and Wenner, 1949; Drozd and Donahue, 2011;



Johnson and Marston, 2008; Khan et al., 2017; Ziemann and Atkinson, 2012). Although these reactive species have been known for decades, it is only recently that the synthesis and measurement of some simple SCIs has become possible (Sheps et al., 2014; Taatjes et al., 2008, 2013; Welz et al., 2012). SCIs can oxidize a series of trace species in the atmosphere, such as $SO_2$, $NO_x$, carboxylic acids, carbonyl compounds, alcohols, etc. (Berndt et al., 2015; Elsamra et al., 2016; Khan et al.,

2018; Mauldin III et al., 2012; Taatjes et al., 2014), and the reaction rate constants reported in the last few years are several orders of magnitude larger than the values estimated earlier (Osborn and Taatjes, 2015; Sander, 2014; Taatjes, 2017), indicating that the contribution of SCIs to atmospheric oxidizing capacity should not be neglected. Reactions of SCIs are also important in secondary organic aerosol (SOA) formation because some bimolecular reactions of SCIs produce low-volatile products (Chhantyal-Pun et al., 2018; Kim et al., 2015; McGillen et al., 2017). In addition, SCIs are reported to

participate in chain reactions with $RO_2$ radicals and carboxylic acids resulting in oligomers formation (Sakamoto et al., 2013; Zhao et al., 2015). The reaction pathways of SCIs in ozonolysis systems are complex, and it is difficult to determine the reaction mechanism of each pathway, especially for those complex SCIs such as monoterpene-derived SCIs. Although the important role of SCIs in aerosol formation has been claimed, a quantitative study of the contribution of SCIs to SOA in alkenes ozonolysis is lacking. Among the SCIs generated, the amount of SCIs that would participate in aerosols formation

and the effects of experimental conditions remain unknown. These questions have not been investigated in detail, yet they are important for promoting our understanding of the fate of SCIs in alkenes ozonolysis.

In the past years, many efforts have been devoted to studying the reaction mechanisms of SCIs. Some matters remain in dispute and one of them is the effect of water. Through synthesizing and measuring SCIs containing less than three carbon atoms, it is found that the reaction of SCIs and water is structure-dependent, while the reaction mechanisms of more complex

SCIs are still unclear because of the lack of synthesis and direct measurements (Huang et al., 2015; Lin and Chao, 2017). In addition to acting as a gas-phase sink of SCIs, water promotes the heterogeneous reactions of SCIs on aqueous surfaces (Enami and Colussi, 2017a; Kumar et al., 2018), which are purported to occur on the surfaces of droplets, aerosols, seas, etc. SCIs reactions at air-water interfaces have drawn much attention in recent years because of the high reactivity of SCIs on aqueous surfaces, especially as applies to the large surface areas present in the atmosphere (Zhong et al., 2018). In the

ozonolysis of alkenes, the presence of water changes the reaction pathways of SCIs, while the yields of products, such as peroxides and carbonyls, formed from SCIs reactions with water have been reported to vary widely (Huang et al., 2013; Li et al., 2016; Berndt et al., 2003). Furthermore, the dependences of these products on relative humidity (RH) have been a point of controversy (Anglada et al, 2002; Hasson et al., 2001, 2003; Ma et al., 2008; Tillmann et al., 2010). In ozonolysis system examining the effects of the addition of compounds that could be oxidized by SCIs, the presence of water had different

effects on the consumption of these added reactants (Newland et al., 2018; Sipilä et al., 2014; Ye et al., 2018). The reactions of SCIs with water further impact the generation of semi-volatile organic compounds (SVOCs) and low-volatile organic compounds (LVOCs), resulting in changes of SOA yield and composition. It is intriguing to note that previous studies presented positive, negative, and neutral effects of water on aerosol formation in alkenes ozonolysis (Bonn et al., 2002;



Bracco et al., 2019; Hessberg et al., 2009; Jonsson et al., 2008; Li et al., 2019; Warren et al., 2009; Yu et al., 2011),
indicating that the uncertainty and complexity of SOA formation as RH changes need further research.

Previous studies have mainly chosen smog chambers as reaction equipment, yet the long reaction time and the large wall loss
may significantly affect the analysis results (Chuang and Donahue, 2017; Zhang et al., 2014, 2015). As reported by Brune et
al. (2019), the Chamber Wall Index, which could be used to evaluate the degree that walls alter the chemistry, demonstrates
that the wall effect in flow reactors is smaller than that in chambers. In this study, experiments were carried out in flow tube
reactors to constrain the reaction time within a few minutes, with the hope of reducing the wall loss effect and allowing
observation of the products generated in the initial state of the reaction. Limonene is selected as the model compound for this
research, not only because of the large emissions of limonene both from biogenic and anthropogenic sources (Andersson-
Sköld and Simpson, 2001; Atkinson and Arey, 2003), but also because of the high SOA formation potential of limonene as a
double-unsaturated terpene (Lee et al., 2006; Ng et al., 2006). This study focuses on quantifying the impact of SCIs on
aerosol formation at different RH, and SCIs scavengers are used to control the amounts of SCIs in the reaction system. In
order to have a complete understanding of limonene-derived SCIs, our research strategy consisted of two sets of experiments,
which separately investigated the first-generation oxidation with endocyclic double bonds (endo-DB) ozonated and the
second-generation oxidation with exocyclic double bonds (exo-DB) ozonated.

## 2 Experimental

### 2.1 Apparatus and procedures

One-stage and two-stage apparatuses were used to investigate SOA formation in the first- and second-generation oxidations,
respectively. Limonene ozonolysis primarily took place on endo-DB, with a rate constant of $2 \times 10^{-16}$ cm$^3$ molecule$^{-1}$ s$^{-1}$
(Atkinson, 1990; Shu and Atkinson, 1994), and the exo-DB reaction with O$_3$ was about 30 times slower (Pathak et al., 2012).
In the first set of experiments focusing on the first-generation oxidation, we used an 8 L quartz flow tube reactor (2 m length,
70 mm inner diameter), which was equipped with a water jacket to control the reaction temperature at $298.0 \pm 0.5$ K. Gas
containing limonene was generated with a diffusion tube kept at a specific temperature, and the concentration of limonene
was detected to be $90.0 \pm 1.5$ ppbv by a gas chromatograph with a flame ionization detector (GC-Agilent 7890A, USA). O$_3$
was generated through photolysis of pure oxygen with a low-pressure Hg lamp, and the concentration of O$_3$ was detected to
be $270.0 \pm 3.0$ ppbv by indigo disulphonate spectrophotometry. To avoid the disturbance by OH radicals, excessive 2-
butanol was generated with a bubbler and added to the reactor as an OH scavenger. The concentration of 2-butanol was
$300.0 \pm 6.0$ ppmv and was estimated to be sufficient for scavenging more than 99% OH radicals. Acetic acid (AA), which
has been considered as an efficient SCIs scavenger due to the rapid reaction of SCIs with AA (Ahmad et al., 2017; Yao et al.,
2014), was used as an SCIs scavenger in this study. Gas containing AA was prepared by an evacuated steel canister (15 L,
Entech Instruments), of which the outlet was linked with a mass flow controller to regulate the gas flow rate. Water vapor
was produced by passing N$_2$ or O$_2$ through a water bubbler, and the RH in reactor ranged from dry condition (< 0.5% RH) to



± 3% RH. Chemicals used in this study were shown in the Supplement. Gases containing limonene, $O_3$, 2-butanol, AA, and water vapor were rapidly blended in a mixing ball and successively introduced into the flow tube reactor. The total gas flow rate was 2 standard L min$^{-1}$ and the residence time was 240 s. In the first set of experiments, endo-DB consumed by $O_3$ was estimated to be ~ 24.6 ppbv, while exo-DB ozonolysis was not taken into account.

In the second set of experiments focusing on the second-generation oxidation, we used a two-stage apparatus including a 2 L quartz flow tube reactor (1 m length, 50 mm inner diameter) and an 8 L quartz flow tube reactor (2 m length, 70 mm inner diameter) in series. The experiments were performed at 298.0 ± 0.5 K in the dark, and the diagram of the reaction equipment is shown in Fig. 1. To investigate the exo-DB oxidation alone a key point was separating the endo-DB reaction, so the major role of the first stage was to accomplish the reaction of endo-DB with $O_3$. Although it was unavoidable that some exo-DB

also reacted with $O_3$ in the first stage, this would not impact the analysis of the second-generation oxidation in the second stage because the quantitative investigation only focused on the exo-DB ozonated in the second-stage. The initial concentrations of limonene and $O_3$ in the first stage were 46.2 ± 1.0 ppbv and 12.2 ± 0.3 ppmv, respectively, and excessive 2-butanol was added to scavenge OH radicals. With a total flow rate of 1.95 standard L min$^{-1}$, the residence time in the first stage was 65 s and the remaining limonene was estimated to be less than 0.4 ppbv. Before entering the second reactor, gas

containing different concentrations of AA was added to mix with the gas out of the first reactor. The total flow rate in the second stage was 2 standard L min$^{-1}$, and it was estimated that the amount of exo-DB that was ozonated was ~ 15.2 ppbv. Assuming a typical 24 h average ambient $O_3$ concentration of 30.0 ppbv (Palm et al., 2018), the equivalent atmospheric times of $O_3$ exposure in the first- and second-generation oxidations were 0.6 h and 26.4 h, respectively.

**2.2 Measurements**

To detect the concentration of AA, gas samples out of the reactor were collected in a glass coil collector rinsed by ultrapure water (18 MΩ) at 277 K. The effective length of the coil collector was ~ 100 cm, in which the flow rates of gas and water were 0.9 standard L min$^{-1}$ and 0.2 mL min$^{-1}$, respectively. The coil collector, shown as a diagram in the Supplement, was as elaborated before (Hua et al., 2008), and the percentage of AA dissolved in the rinsing solution was calculated to be higher than 99% based on Henry's law when the partitioning between gas phase and liquid phase was at equilibrium (Lazrus et al.,

1986). Samples of extracting solution were analyzed by an ion chromatography (IC, DIONEX ICS-2000) with the detection limit of ~ 50 pptv, and the standard solution of AA was prepared to perform calibrations in every measurement. Wall loss experiments were carried out by introducing gas containing AA to the reactor at 2 standard L min$^{-1}$, and the inlet and outlet concentrations of AA at different RH were measured. The loss fraction of AA in the 8 L flow tube was found to be less than 8% without distinct influence by RH, and the results reported below were rectified by the wall loss effect. In both of the first-

and second-generation oxidations, when no AA was added, the background level of AA was detected to be less than 1.0 ppbv, which was subtracted from the results.



Particle size distribution was measured by a scanning Mobility Particle Sizer (SMPS, 3938, TSI) consisting of a differential mobility analyzer (DMA, 3081A) and a condensation particle counter (CPC, 3776). The measurement was in the range of 13.8–504.8 nm, and the interval was 5 min. The sampling flow rate of SMPS was 0.3 L min$^{-1}$ and the sheath air flow was 3 L min$^{-1}$. All data were treated with multiple charge correction and diffusion loss correction by the TSI software, and the aerosol density was assumed to be 1.3 g cm$^{-3}$ when calculating the aerosol mass concentration (Saathoff et al., 2009; Wilson et al., 2015). The gas flow in the 8 L flow reactor was evaluated to be laminar as the Reynolds number was ~ 43 (Ezell et al., 2010), and to better observe the process of SOA formation during the reaction, a stainless tube, which was flexible in the axial direction, was used to collect samples at different positions in the reactor. As the gas sample extracted from the stainless steel tube was a small part of the total gas flow, the whole flow state in the reactor could remain stable. The gas samples collected at different positions represented products formed under different reaction times. The measurement of the SOA formation process in the 8 L flow reactor started after the reaction had proceeded for 60 s, and the sampling interval was set as 30 s. It was noted that this method was used to assist us to know the processes of SOA formation in the first- and second-generation oxidation. The calculations and analysis in this study used the samples collected at the end of the reactor. The aerosol wall loss experiments were performed as our previous study (Gong et al., 2018), and the loss fraction of SOA mass concentration was found to increase when RH increased as shown in Table S1. The mass concentration of SOA was also subtracted by the water content according to the hygroscopic growth factor reported before (Bateman et al., 2015).

### 3 Results and discussion

#### 3.1 Regulating and quantifying the amount of SCIs

To quantitatively investigate the performance of SCIs in SOA formation, the amount of SCIs in the reaction system was regulated by adding different concentrations of AA (24–480 ppbv) at a series of RH (< 0.5%, 10%, 40%, 60%, 80%). Both of AA and water could react with SCIs, and here the amount of SCIs in reaction system is defined as the accumulated amount of SCIs that are not consumed by AA and water, and it could be derived by deducting the amount of SCIs reacted with AA and water from the amount of total SCIs generated. Limonene ozonolysis generates several kinds of SCIs (Leungsakul et al., 2005), which are difficult to distinguish. As reported before, SCIs reaction with water is structure-dependent, and here all SCIs are divided into two types, one of which tends to react with water (SCI$_\text{I}$), and the other is inert to react with water (SCI$_\text{II}$) (Long et al., 2018). The molar yield of SCIs, defined as the ratio of SCIs mole number to the mole number of DB ozonated, could be inferred through measuring the consumption of AA ($\Delta$AA). This calculation was based on two points: first, previous studies reported that SCIs reaction with AA was rapid and not structure-dependent (Khan et al., 2018); second, as we described below, the SCIs yield derived from reaction with AA was higher that derived from reaction with water, confirming that both kinds of limonene-derived SCIs reacted effectively with AA. With the increase of AA concentration, $\Delta$AA increased in the beginning and then became stable. According to the maximum $\Delta$AA under dry conditions, the SCIs yield of endo-DB ozonated was calculated to be ~ 0.44. Through measuring the generation of H$_2$O$_2$, the



yield of SCI$_I$ in endo-DB ozonated was reported to be ~ 0.24 (Gong et al., 2018), and thus the yield of SCI$_{II}$ was derived to be ~ 0.20.

For the amount of SCIs consumed by SCIs scavengers, ΔAA represented the amount of SCIs consumed by AA, and the amount of SCIs consumed by water was calculated through the reaction rate ratio of SCIs reaction with water and AA, which was elaborated in the Supplement. The ratio of rate constants of SCIs reaction with water and AA was derived from the estimation of ΔAA as shown in Fig. 2. Due to the competition with water, the variation of ΔAA under high-humidity

conditions was gentler than that under dry conditions. The relationship between ΔAA and the concentration of AA is estimated according to Eqs. (1) to (3) (Bracco et al., 2019; Hessberg et al., 2009):

$$\frac{SCI_{AA}}{SCI_{Total}} = \frac{\Delta AA}{SCI_{Total}} = \frac{k_{(SCI+AA)} \cdot [AA]}{k_{(SCI+AA)} \cdot [AA] + k_{(SCI+H2O)} \cdot [H_2O] + k_{(other)}} \tag{1}$$

$$\Delta AA = \frac{1}{1 + \dfrac{k_{(SCI+H2O)} \cdot [H_2O] + k_{(other)}}{k_{(SCI+AA)} \cdot [AA]}} \cdot SCI_{Total} \tag{2}$$

$$\Delta AA = \frac{SCI_{I}}{1 + \dfrac{k_{(SCI+H2O)I} \cdot [H_2O] + k_{(other)I}}{k_{(SCI+AA)} \cdot [AA]}} + \frac{SCI_{II}}{1 + \dfrac{k_{(other)II}}{k_{(SCI+AA)} \cdot [AA]}} \tag{3}$$

where SCI$_{AA}$ (molecule cm$^{-3}$) is the amount of SCIs reacted with AA; SCI$_{Total}$ (molecule cm$^{-3}$) is the amount of total SCIs generated; ΔAA (molecule cm$^{-3}$) is the amount of AA consumed; $k_{(SCI+AA)}$ (cm$^3$ molecule$^{-1}$ s$^{-1}$) is the rate constant of SCIs reaction with AA; [AA] (molecule cm$^{-3}$) is the concentration of AA; $k_{(SCI+H2O)}$ (cm$^3$ molecule$^{-1}$ s$^{-1}$) is the rate constant of SCIs reaction with H$_2$O; [H$_2$O] (molecule cm$^{-3}$) is the concentration of H$_2$O; $k_{(other)} = k_{(isomerization)} + k_{(SCI+products)} \cdot$ [products], meaning that $k_{(other)}$ accounts for the sum of SCIs isomerization and reaction with other products in the system. Equation (2)

could be extended to Eq. (3) containing two types of SCIs. One consideration in calculations is whether the reaction with H$_2$O or with (H$_2$O)$_2$ dominates in SCI$_I$ reaction with water, which is discussed in the Supplement. Results showed that the reaction with H$_2$O was more important in this reaction system, and the values of $k_{(SCI+H2O)I}$ and $k_{(other)I}$ were derived to be 5 × 10$^{-16}$ cm$^3$ molecule$^{-1}$ s$^{-1}$ and 30 s$^{-1}$ (Fig. S2). Estimations of ΔAA at < 0.5% and 80% RH were calculated and shown in Fig. 2, where $k_{(SCI+AA)}$ and $k_{(other)II}$ were derived to be 1 × 10$^{-10}$ cm$^3$ molecule$^{-1}$ s$^{-1}$ and 100 s$^{-1}$, respectively. The ratio of rate

constants of SCIs reaction with water and AA was 5 × 10$^{-6}$ and the amounts of SCIs under different conditions were shown in Table S2.

In the second-generation oxidation, according to the maximum ΔAA under dry conditions, the SCIs yield of exo-DB ozonated was calculated to be ~ 0.60. As RH increased to 40%, ΔAA became smaller than that under dry conditions because of the competition of water. However, at 60% RH, ΔAA performed a rise and at 80% RH the maximum of ΔAA increased to

be about double of that under dry conditions, suggesting that more SCIs were generated due to more exo-DB ozonated. The amount of exo-DB ozonated under high-humidity conditions could be derived through estimating the variation of ΔAA.



Through calculating the formation of $H_2O_2$ as described above, the yields of $SCI_I$ and $SCI_{II}$ in the second-generation oxidation were both assumed to be 0.30. According to the calculation results (Fig. S3), the amounts of exo-DB ozonated at 60% and 80% RH were 1.3 and 2.0 times of that under dry conditions. If particles were treated as liquid state under high-humidity conditions, the amount of exo-DB reaction with $O_3$ in aerosols was calculated in the Supplement. Bulk-phase ozonolysis could not explain the exo-DB ozonated, suggesting that reactions on surfaces were more important. Some studies reported the uptake and ozonation of terpenes on aqueous surfaces (Enami et al., 2010; Matsuoka et al., 2017), and as for those semi-volatile products containing exo-DB, their uptake on aqueous surfaces might have a greater chance to happen due to the lower volatility. This study reported direct evidence for the heterogeneous oxidation of exo-DB of limonene, which was proposed before by Zhang et al. (2006). The uptake coefficient of these unsaturated products was estimated to be $10^{-3}$ of magnitude, which was elaborated in the Supplement. The amounts of SCIs in the second stage under different conditions were calculated and appeared in Table S3.

### 3.2 The effect of water on SOA formation

### 3.2.1 Water effect with the presence of SCIs

In ozonolysis water could participate in some reactions and it is necessary to investigate SOA formation under different RH. When investigating the SOA formation process experiments were carried out from < 0.5% RH to 90% RH with the interval of 10% RH. Through calculating the amounts of limonene reacted at different reaction times, the growth curves of SOA mass concentration and SOA yield could be derived in the first-generation oxidation, as shown in Fig. S4. SOA yield was determined from the mass concentration of SOA divided by the mass concentration of limonene reacted. As RH increased, SOA formation presented non-monotonic dependence on RH. The presence of water suppressed the process of SOA formation under low-humidity conditions (10–50% RH), while the inhibitory effect of water on SOA formation was not observed under high-humidity conditions (60–90% RH). The SOA yields at the end of the first-generation oxidation under dry and high-humidity conditions were ~ 20%, and under low-humidity conditions the SOA yield was ~ 15%. In the second-generation oxidation, SOA growth was observed due to exo-DB ozonolysis, and the increase of SOA mass concentration (ΔSOA) was derived by subtracting the SOA mass concentration at the end of the first stage from that at the end of the second stage. Because limonene was almost consumed in the first stage, the variations of SOA mass concentration and SOA yield in the second stage are shown as a function of reaction time (Fig. S5). The inhibitory effect of water on SOA formation under low-humidity conditions was also observed with a ΔSOA reduction of as much as ~ 22%. Under high-humidity conditions, SOA growth became more intense due to more exo-DB ozonated, and the increment of SOA yield above 80% RH was more than twice that under dry conditions.

### 3.2.2 Water effect without the presence of SCIs

To figure out the fraction of SCIs reactions contributing to SOA formation, excess AA (10.0 ± 0.4 ppmv) was added to scavenge all SCIs in the reaction system. The concentration of AA used here was estimated to be sufficient for scavenging


more than 99% of SCIs generated during reactions as claimed in the Supplement. Without the presence of SCIs, the amount
of SOA decreased dramatically both in the first- and second-generation oxidation, indicating that the reaction with AA
converted SCIs into more volatile products. The mechanisms of SCIs reaction with carboxylic acids were not entirely clear,
and two reaction pathways were proposed. The insertion products, produced from hydroperoxyester channel, would go
through water elimination and formed acid anhydride (Aplincourt and Ruiz-López, 2000; Cabezas and Endo, 2020; Long et
al., 2009). As the molecular weight of AA was small, the contribution of acid anhydride produced to SOA might be limited.
Some studies suggested that SCIs reaction with acids proceeded via acid-catalyzed tautomerization of SCIs to vinyl
hydroperoxides, which were not likely to contribute to aerosols (Kumar et al., 2014a; Liu et al., 2015). Besides, it was
observed that when scavenging the same amount of SCIs by AA and water the decreases of SOA were similar. Thus, it was
estimated that the contribution of the products formed from SCIs reaction with AA to aerosols was small. Through excluding
the impact of SCIs, it was found that SCIs reactions accounted for ~ 63% and ~ 73% of the total SOA formed under dry
conditions in the first- and second-generation oxidation.

Figure 3 shows how the presence of SCIs impacts SOA formation, and the influence of water on SOA formation with the
presence of SCIs was different from that without the presence of SCIs. The inhibitory effect of water on SOA formation
could be attributed to the reaction between SCIs and water, producing α-hydroxyalkyl hydroperoxides, which were thought
to preferentially decompose to $H_2O_2$ and aldehydes (Chen et al., 2016; Jiang et al., 2013; Kumar et al., 2014b; Winterhalter
et al., 2000). The aldehydes, such as pinonaldehyde and nopinone formed from α-pinene-derived and β-pinene-derived SCIs
reaction with water, displayed higher volatility and their contribution to aerosols was reported to be small in ozonolysis
(Emanuelsson et al., 2013; Fick et al., 2003; Jenkin, 2004; Mutzel et al., 2016; Sakamoto et al., 2017). An intriguing finding
here was that the amount of SOA formed under high-humidity conditions increased, and one possible reason for this
phenomenon was the physical effect of water, as the viscosity of particles would decrease due to the uptake of water
resulting in the transition of particles from non-liquid to liquid state with increasing RH (Faust et al., 2017; Renbaum-Wolff
et al., 2013). The low viscosity of particles further improved the particle-phase diffusion and gas-particle partitioning of
SVOCs (Shiraiwa and Seinfeld, 2012; Ye et al., 2016). Nevertheless, some studies reported that the SOA formation process
in monoterpene ozonolysis was quasi-equilibrium growth, meaning that the timescales of gas-particle partitioning
equilibrium and diffusion in particles were much smaller than the timescales of gas-phase reactions and the wall loss process
(McVay et al., 2014, 2016; Nah et al., 2016, 2017; Riipinen et al., 2011). Results showed that when SCIs were scavenged in
the first-generation oxidation, the amount of SOA was almost unaffected by RH. Besides, considering that the diameters of
most particles generated in the experiments were smaller than 100 nm, and the effect of water on the diffusion limit was not
considered as the major cause of the increase of SOA under high-humidity conditions (Tu and Johnson, 2017; Veghte et al.,
2013). Figure 4 shows the dependence of the fraction of SCIs contribution to SOA on RH, and it was found that in both of
the first- and second-generation oxidations the contribution of SCIs reactions to SOA formation decreased under low-



humidity conditions and increased under high-humidity conditions, suggesting that the non-monotonic effect of water on SOA formation was relevant to SCIs reactions.

## 3.3 The correlation between SCIs and SOA formation

### 3.3.1 Quantifying the fraction of SCIs converting into SOA

The variation of SOA with the concentration of AA at different RH in the first-generation oxidation was shown in Fig. 5. Although the amount of SOA formed under dry and high-humidity conditions were similar, their variations were different with increasing AA, indicating that differences existed in SOA formation mechanisms between dry and high-humidity conditions. To figure out the correlation between SCIs and SOA formation, the dependence of SOA mass concentration on the amount of SCIs is shown in Fig. 6, where the dependences under dry and low-humidity conditions are similar to each

other with the deviation among their slopes of linear fitting lines falling within 15%. The fitting lines under high-humidity conditions are also similar. The fact that the correlation coefficients ($R^2$ values) of the fitting lines are greater than 0.9 indicates that the amount of SOA generated has a significant positive correlation with the amount of SCIs, regardless of RH and oxidation degree. In the first-generation oxidation, the slope of the fitting line under dry and low-humidity conditions is $6.26 \times 10^{-11}$, and the slope under high-humidity conditions is $1.05 \times 10^{-10}$. In the second-generation oxidation, the slope of

the fitting line under dry and low-humidity conditions is $5.08 \times 10^{-11}$, and the slope of the fitting line at 60% RH is $9.43 \times 10^{-11}$, which is similar to the value of $1.08 \times 10^{-10}$ at 80% RH. The slopes of fitting lines in the first- and second-generation oxidations under dry and low-humidity conditions were similar, as were the fitting results under high-humidity conditions, suggesting that the role of SCIs in SOA formation was slightly affected by the oxidation degree in limonene ozonolysis. Based on the correlation between SOA formation and SCIs, the SOA yield of limonene-derived SCIs could be inferred from

the mass concentration of SOA formed from SCIs reactions divided by the mass concentration of SCIs reacted. The SOA yield of limonene-derived SCIs was estimated to be ~ 0.20 under dry and low-humidity conditions and ~ 0.35 under high-humidity conditions. From another view, results showed that among the SCIs in reaction system, the fraction that could produce low-volatile products and convert into SOA was stable, and here the fraction was denoted as $\alpha_{SCI}$. In this study $\alpha_{SCI}$ of limonene-derived SCIs was estimated to be 11–17% under dry and low-humidity conditions and 20–30% under high-

humidity conditions, which was elaborated in the Supplement. Considering that part of products in particles were semi-volatile, $\alpha_{SCI}$ derived here was expected to be a lower limit, and this value was used to estimate the amount of SOA formed from SCIs reactions.

### 3.3.2 Analysis on reaction mechanisms

The SOA formation potential of SCIs under high-humidity conditions was found to be more significant than that under dry

and low-humidity conditions, which needed explanations on reaction mechanisms. In ozonolysis the cycloaddition of $O_3$ to alkenes produced a primary ozonide (POZ), which decomposed to ECIs and aldehyde or ketone. ECIs can isomerize through hydroperoxide channel, rearrange to esters, or stabilize to form SCIs (Johnson and Marston, 2008). In this study, SOA



formation was roughly divided into two pathways: directly from ECIs reactions and from SCIs reactions. According to the results, SOA formation directly from ECIs reactions was unaffected by changing RH. As for SCIs, the main reaction

pathways concluded bimolecular reactions with products formed during ozonolysis (Lee and Kamens, 2005; Yao et al., 2014), unimolecular reactions producing vinyl-hydroperoxides, secondary ozonides, and dioxiranes, etc. (Long et al., 2019), and chain reactions producing oligomers (Sakamoto et al., 2013). The reaction scheme of limonene ozonolysis and the structures of proposed products were shown as Fig. 7. When discussing the behaviors of SCIs in SOA formation, side reactions of SCIs needed consideration and were analyzed below.

Since the concentration of $O_3$ in experiments was higher than the concentration of limonene, the reaction of SCIs with $O_3$ might impact products formation. A lower limit for the rate constant of SCIs reaction with $O_3$ of $10^{-18}$ $cm^3$ $molecule^{-1}$ $s^{-1}$ was proposed at 298 K (Kjaergaard et al., 2013), and some studies derived a higher value of about $10^{-14}$ $cm^3$ $molecule^{-1}$ $s^{-1}$ (Chang et al., 2018; Vereecken et al., 2015). The reaction of limonene-derived SCIs and $O_3$ was rarely reported, and we took the rate constant of $10^{-14}$ $cm^3$ $molecule^{-1}$ $s^{-1}$ to estimate an upper limit for the amount of SCIs reacted with $O_3$. Taking SCIs

reaction with AA as a reference, when using the lowest concentration of AA in experiments, the ratios of the amount of SCIs reacted with $O_3$ to the amount of SCIs reacted with AA were 0.001 and 0.046 in the first- and second-generation oxidation, respectively, indicating that the reaction of SCIs with $O_3$ was not important. SCIs could react with alcohols, and here the effect of this reaction was considered because in this study high concentration of 2-butanol was used to scavenge OH radicals. The rate constants of $CH_2OO$ reaction with methanol and ethanol were measured to be about $10^{-15}$ $cm^3$ $molecule^{-1}$

$s^{-1}$ at 298 K (McGillen et al., 2017). Theoretical computation showed that the rate constants of $CH_2OO$ and $(CH_3)_2COO$ reaction with methanol at 298 K were about $10^{-13}$ $cm^3$ $molecule^{-1}$ $s^{-1}$ and $10^{-15}$ $cm^3$ $molecule^{-1}$ $s^{-1}$, respectively (Aroeira et al., 2019). When the rate constant of limonene-derived SCIs reaction with 2-butanol was assumed as $10^{-15}$ $cm^3$ $molecule^{-1}$ $s^{-1}$, it was estimated that with the lowest concentration of AA used in experiments, the ratio of the amount of SCIs reacted with 2-butanol to the amount of SCIs reacted with AA was 0.125, which was considered to be not important enough to

influence the results. Besides, products formed from SCIs reaction with AA and water might further react with SCIs and impacted aerosols formation (Chen et al., 2019). However, due to the short lifetime and low concentration of SCIs in system, the concentration of SCIs was a limiting factor in bimolecular reactions of SCIs with other products. The reactions of SCIs with the compounds formed from SCIs scavengers would not compensate the effect of the consumption of SCIs on SOA formation.

**3.3.3 Reasons for different performances of SCIs under different RH**

As a turning point of RH in SOA formation potential of SCIs was observed, which could not be explained by gas-phase reaction mechanisms, the different performances of SCIs between low- and high-humidity conditions were speculated to be due to the impact of the change of aerosol phase state on some chemical reactions. Some studies reported that the transition for the state of monoterpene-derived SOA was 65−90% RH (Bateman et al., 2015, 2016), while the transition for SOA

chemical reactivity was 35−45% RH (Li et al., 2015). A similar turning point of RH in the formation of particulate dimers in



monoterpene ozonolysis was reported by Kristensen et al. (2014), who observed that the concentration of dimers in particles above 50% RH was double that observed below 50% RH. Heterogeneous reactions on aqueous aerosols impacted the formation of SOA (Knote et al., 2014; Woo et al., 2013), and here two reaction pathways were proposed. Liquid surfaces have been proven to confine SCIs into a specific orientation, which helps to maintain the stability of SCIs and provide more opportunities for SCIs to react with other species (Qiu et al., 2018a; Zhong et al., 2017). Previous studies reported that SCIs could react with a series of compounds at air-water interfaces (Heine et al., 2018; Kumar et al., 2017, 2019; Qiu et al., 2018b; Xiao et al., 2018), and some low-volatile products were observed (Enami and Colussi, 2017b, c). Under high-humidity conditions, the heterogeneous reactions of SCIs might exist with other reaction pathways of SCIs and improved the SOA formation potential of SCIs. In addition, a significant amount of $H_2O_2$ formed from SCIs reaction with water was observed in ozonolysis under high-humidity conditions (Jiang et al., 2013; Qiu et al., 2019), which might impact the aerosol-phase chemistry. $H_2O_2$ was reported to play an important role in the nonradical oxidation of carbonyls in aqueous phase (Galloway et al., 2011; Herrmann et al., 2015), producing hydroxyhydroperoxides and promoting SOA formation (Zhao et al., 2012, 2013). The $H_2O_2$ oxidation of carbonyls was speculated to mainly occurred in surface liquid layer of aerosols, resulting in the generation of organic peroxides and high-molecular-weight oligomers (Sui et al., 2017; Zhang et al., 2019). The impact of $H_2O_2$ reactions at air-liquid interface might be another reason for the performances of SCIs in SOA formation under high-humidity conditions. The inhibitory effect of water on aerosol formation in monoterpene ozonolysis was also reported by Li et al. (2019), yet they found that the $RO_2$-derived highly oxidized molecules (HOMs) formation was not influenced by RH, and the reason was expected to be the SCIs-derived HOMs. The main expectation of this study was to quantify the role of SCIs in SOA formation, and a limitation was the lack of measurement of aerosol composition. In ozonolysis system the detection of products formed from SCIs reactions was complicated because SCIs could react with multiple products formed from ozonolysis, and the definite reason for different SOA formation potentials of SCIs between low- and high-humidity conditions perhaps needed more composition measurements.

### 3.4 The contribution of limonene-derived SCIs to SOA in the atmosphere

Due to the bimolecular reactions of SCIs, SOA formation from limonene ozonolysis could be influenced by other species in the atmosphere, and to evaluate the contribution of limonene ozonolysis to SOA explicitly it is necessary to estimate the contribution of SCIs to SOA. Here three situations, forest, urban area, and indoor area, were analyzed at different RH and the concentrations of reactants were discussed as below. The concentration of limonene in tropical rainforest was reported to be about 0.18 ppbv, and in urban area 0.15 ppbv was taken as an example (Chen et al., 2010; Jia et al., 2008; Yáñez-Serrano et al., 2018). In indoor area, some studies observed high indoor limonene concentration to exceed 80 ppbv (Brown et al., 1994; Li et al., 2002), while a typical indoor concentration of limonene was considered as 2 ppbv (Mandin et al., 2017; Weschler and Carslaw, 2018). The concentrations of $O_3$ in forest and urban area were taken as 45 ppbv (Lelieveld et al., 2008; Williams et al., 2016). Weschler (2000) claimed that the $O_3$ concentration in indoor environments was 20–70% of the outdoor concentration, and 20 ppbv of $O_3$ was taken as a typical indoor concentration (Weschler, 2000; Weschler and


Carslaw, 2018). The concentrations of some other compounds that could impact SCIs reactions, such as $SO_2$ and $NO_2$, were showed in Table S4.

To estimate the contribution of limonene-derived SCIs to SOA in the atmosphere, the concentration of SCIs needed to be calculated and the steady-state approximation was applied as Eq. (4) (Percival et al., 2013):

$$[SCI]_{ss} = \frac{k_{ozo} \cdot [O_3] \cdot [limonene] \cdot Y_{SCI}}{k_{(SCI+H2O)} \cdot [H_2O] + k_{(SCI+SO2)} \cdot [SO_2] + k_{(SCI+NO2)} \cdot [NO_2] + k_{(other)}} \tag{4}$$

where $[SCI]_{ss}$ (molecule $cm^{-3}$) is the steady-state concentration of SCIs; $k_{ozo}$ ($cm^3$ $molecule^{-1}$ $s^{-1}$) is the rate constant of limonene reaction with $O_3$; $[O_3]$ (molecule $cm^{-3}$) is the concentration of $O_3$; [limonene] (molecule $cm^{-3}$) is the concentration of limonene; $Y_{SCI}$ is the molar yield of SCIs; $k_{(SCI+H2O)}$ ($cm^3$ $molecule^{-1}$ $s^{-1}$) is the rate constant of SCIs reaction with $H_2O$, which is $5 \times 10^{-16}$ $cm^3$ $molecule^{-1}$ $s^{-1}$ as derived in this study; $[H_2O]$ (molecule $cm^{-3}$) is the concentration of $H_2O$; $k_{(SCI+SO2)}$ ($cm^3$ $molecule^{-1}$ $s^{-1}$) is the rate constant of SCIs reaction with $SO_2$, which is assumed as $1 \times 10^{-11}$ $cm^3$ $molecule^{-1}$ $s^{-1}$ (Lin and Chao, 2017); $[SO_2]$ (molecule $cm^{-3}$) is the concentration of $SO_2$; $k_{(SCI+NO2)}$ ($cm^3$ $molecule^{-1}$ $s^{-1}$) is the rate constant of SCIs reaction with $NO_2$, which is assumed as $1 \times 10^{-12}$ $cm^3$ $molecule^{-1}$ $s^{-1}$ (Lin and Chao, 2017); $[NO_2]$ (molecule $cm^{-3}$) is the concentration of $NO_2$; $k_{(other)}$ ($s^{-1}$) accounts for the sum of SCIs isomerization and reaction with other VOCs, and the value is also derived in the experiments. The concentrations of $SCI_I$ and $SCI_{II}$ were calculated seperately, and the ozonolysis of both endo-DB and exo-DB were considered. The concentrantion of SCIs was enormously affected by changing RH, and with the increase of RH to 100 % the SCIs concentration decreased to the minimum. In forest, the range of SCIs concentration varies in $2.65 \times 10^3 - 1.01 \times 10^4$ molecule $cm^{-3}$, and in urban and indoor area the ranges of variation are $2.15 \times 10^3 - 7.86 \times 10^3$ molecule $cm^{-3}$ and $1.28 \times 10^4 - 4.65 \times 10^4$ molecule $cm^{-3}$, respectively. According to the SOA formation potential of SCIs, it is estimated that in all of three situations the contribution of SCIs to SOA formation is the lowest under low-humidity conditions, where the contribution of SCIs to SOA is $(5.26 \pm 0.58) \times 10^{-2}$ µg $m^{-3}$ $h^{-1}$ in forest, $(4.26 \pm 0.46) \times 10^{-2}$ µg $m^{-3}$ $h^{-1}$ in urban area, and $(2.52 \pm 0.28) \times 10^{-1}$ µg $m^{-3}$ $h^{-1}$ in indoor area. Under high-humidity conditions, the contribution of SCIs to SOA is $(8.21 \pm 0.15) \times 10^{-2}$ µg $m^{-3}$ $h^{-1}$ in forest, $(6.66 \pm 0.12) \times 10^{-2}$ µg $m^{-3}$ $h^{-1}$ in urban area, and $(3.95 \pm 0.72) \times 10^{-1}$ µg $m^{-3}$ $h^{-1}$ in indoor area, which are similar to those under dry conditions.

**4 Conclusions**

This study investigated SOA formation of both the first- and second-generation oxidations in limonene ozonolysis at different RH, with the aim of extending our understanding of SCIs performances in SOA formation. The reaction pathways of SCIs in ozonolysis system were complex, and to figure out the role of SCIs in SOA formation, the amount of SCIs was regulated to observe the influence on SOA formation and estimate the fraction of SCIs participating in SOA formation. Results showed that SOA formation from ECIs isomerization was scarcely influenced by water, while SCIs reactions helped to explain the uncertainty of water effect on SOA formation in ozonolysis. Although part of SCIs were consumed by water, the contribution of SCIs to SOA under high-humidity conditions still resembled that obtained under dry conditions. The



significant positive correlation between SOA formation and SCIs implied that the SOA formation potential of SCIs could maintain stable without obvious impact from the oxidation degree. The SOA formation potential of SCIs under high-humidity conditions was nearly double that under dry and low-humidity conditions. SOA formation from limonene ozonolysis was impacted by changing RH due to SCIs reactions, and the contribution of SCIs to SOA was needed to be considered in models even under high RH. As an important monoterpene in the atmosphere, limonene has similarities in

chemical structures with its isomers, α-pinene and β-pinene. Limonene owns a similar endo-DB with α-pinene, and a similar exo-DB with β-pinene. The role of limonene-derived SCIs in SOA formation observed here might be compared with α-pinene-derived and β-pinene-derived SCIs and helped to explain the effect of water on SOA formation in ozonolysis, which was a disputable issue. In previous studies, the water effect on SOA formation in α-pinene ozonolysis was reported to be not uniform (Bonn et al., 2002; Jonsson et al., 2008; Li et al., 2019). In β-pinene ozonolysis, water was found to reduce aerosols

formation under low-humidity conditions (Bonn et al., 2002; Emanuelsson et al., 2013). In this study, the performances of SCIs formed from endo-DB and exo-DB ozonolysis of limonene showed consistency. In both of the endo-DB and exo-DB oxidations, water was found to be an uncertainty on SCIs reactions and producing aerosols. On the one hand, gas-phase water consumed part of SCIs and hindered SOA formation; on the other hand, condensed-phase water improved the contribution of SCIs reactions to SOA through changing the phase of aerosols and heterogeneous reactions. This study

provides new insights into the role of SCIs in SOA formation from the quantitative point of view, and further studies on the high-molecular-weight products formed from SCIs are required to comprehend the role of SCIs played in the atmosphere.

*Data availability.* The data accessible by contacting the corresponding author (zmchen@pku.edu.cn).

*Author contributions.* YG designed the study, carried out the experiments, and wrote the paper. ZC helped interpret the results and modified the paper.

*Competing interests.* The authors declare that they have no conflict of interest.

*Acknowledgments.* The authors gratefully thank the National Key Research and Development Program of China (grant 2016YFC0202704) and the National Natural Science Foundation of China (grant 21477002) for financial support.



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





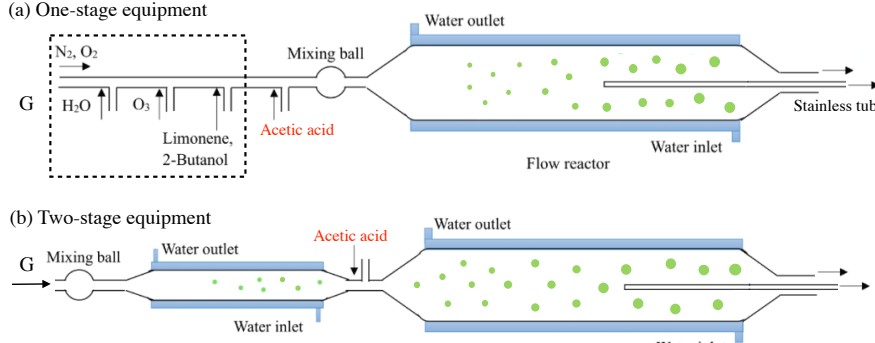

**Figure 1: Diagram of the experimental setup. The (a) one-stage and (b) two-stage apparatuses are used to investigate the first- and second-generation oxidation. The green circles represent the process of SOA growth.**





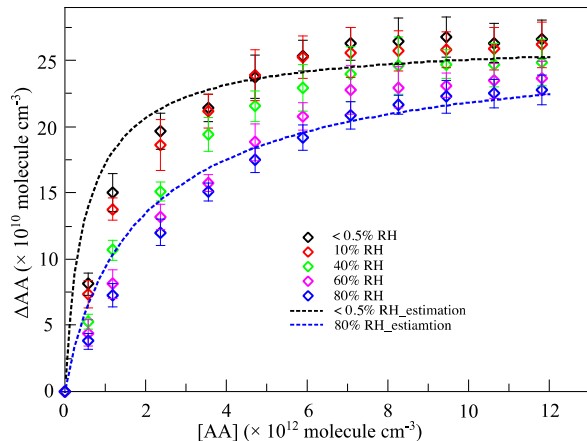

**Figure 2: The variations of the consumption of acetic acid (ΔAA) with the concentration of acetic acid ([AA]) at different relative humidity (RH) in the first-generation oxidation. Scatters: measured ΔAA; Black line: estimated ΔAA at < 0.5% RH; Blue line: estimated ΔAA at 80% RH.**





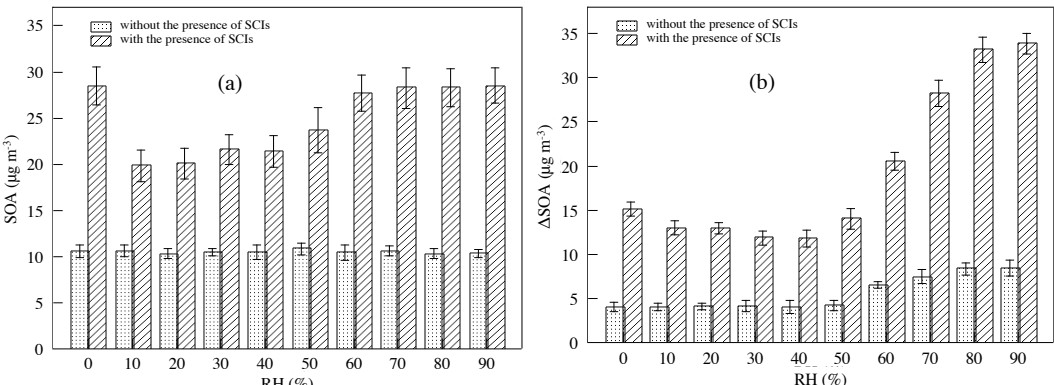

**Figure 3: Impact of the presence of SCIs on (a) SOA mass concentration (SOA) in the first-generation oxidation and (b) the**
**increment of SOA mass concentration (ΔSOA) in the second-generation oxidation at different relative humidity (RH).**





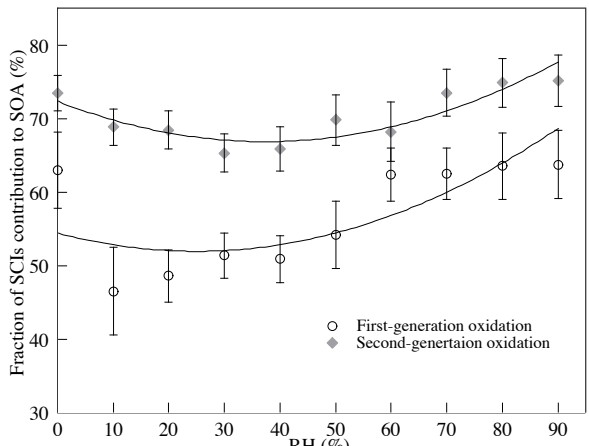

**Figure 4: The fraction of SCIs reactions contributing to SOA formation in the (a) first- and (b) second-generation oxidation at different relative humidity (RH).**




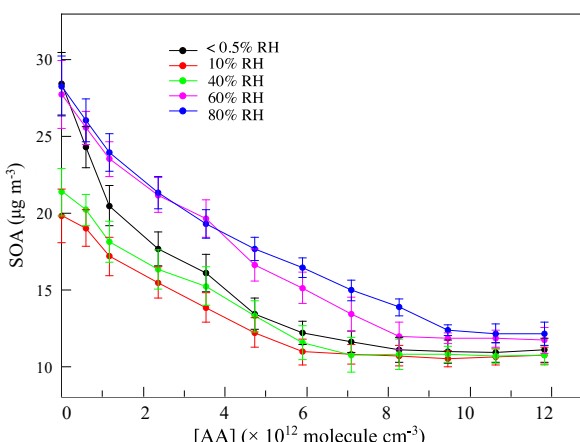

**Figure 5: The variation of SOA mass concentration (SOA) with the concentration of acetic acid ([AA]) at different relative humidity (RH) in the first-generation oxidation.**





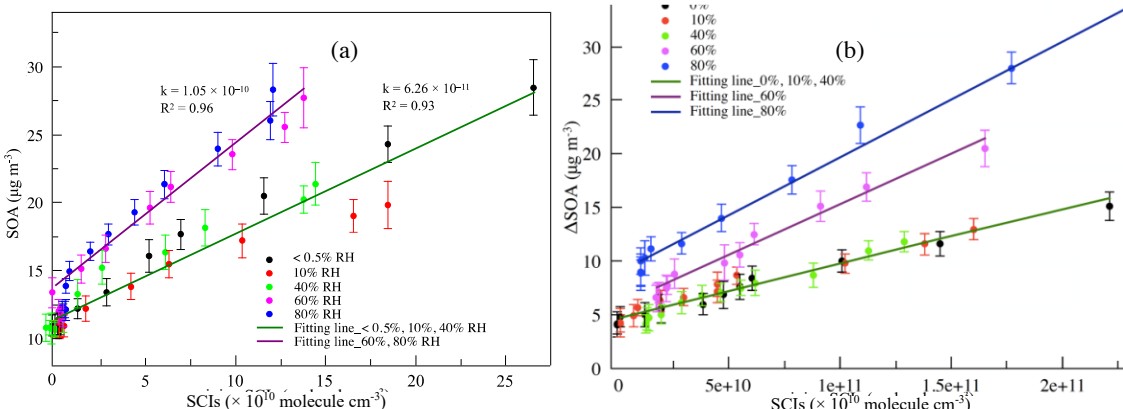


**Figure 6: (a) The dependence of SOA mass concentration (SOA) on the amount of SCIs at different relative humidity (RH) in the first-generation oxidation. (b) The dependence of SOA mass concentration increment (ΔSOA) on the amount of SCIs at different relative humidity (RH) in the second-generation oxidation.**



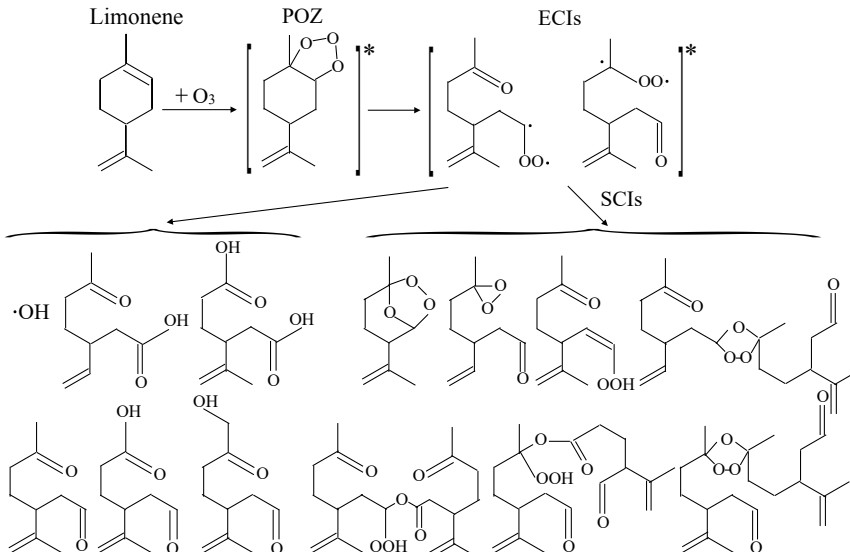


**Figure 7: Reaction scheme of limonene ozonolysis.**