# Peer review of "Quantification of the role of stabilized Criegee intermediates in the formation of aerosols in limonene ozonolysis"

_Atmospheric Chemistry and Physics, 2020_

## Referee Comment (RC1) · Anonymous Referee #1 · 26 Aug 2020

General Comments: Gong and Chen report the experimental study on the formation of SOA during the ozonolysis of limonene, a class of important biogenic VOC in the atmosphere. They used flow tube reactors under different relative humidity (RH) to investigate the mechanism of SOA formation, especially the role of stabilized Criegee intermediates (SCIs). Their findings imply the different mechanisms of SOA formation at dry vs humid condition. The subject is within the scope of ACP and some findings seem important from the viewpoint of atmospheric aerosol chemistry. However, I am concerned about some critical issues that should be addressed.

Specific Comments: 1. I am most concerned about the side reaction of SCIs with 2-

butanol. In page 10, the authors wrote "The rate constants of CH2OO reaction with methanol and ethanol were measured to be about 10^−15 cm3 molecule−1 s−1 at 298 K (McGillen et al., 2017)", but this statement is incorrect. Actually, MeGillen et al. (ACS Earth Space Chem. 2017, 1, 664−672) experimentally determined the rate constants k ∼ 10^-13 cm3 molecule−1 s−1 for CH2OO + CH3OH/C2H5OH and k ∼ 4 x 10^-14 cm3 molecule−1 s−1 for (CH3)2COO + CH3OH at ∼ 298 K. Furthermore, Tadayon et al. (J. Phys. Chem. A 2018, 122, 1, 258–268) reported the rate constants of (1.9 ± 0.5) × 10^−13 cm3 molecule-1 s–1 for the reaction of CH2OO with 2-propanol at 295 K. Hence, the assumption that the rate constant of limonene-derived SCIs reaction with 2-butanol is 10^−15 cm3 molecule−1 s−1 seems to be inadequate. If the authors assumed the rate constant to be 10^−13 cm3 molecule−1 s−1, then the ratio of the amount of SCIs reacted with 2-butanol to the amount of SCIs reacted with AA would be much larger than the value authors claimed. Thus, SCIs in the presence of an excess amount of 2-butanol would be exclusively converted into alpha-alkoxyalkyl-hydroperoxides, that may contribute to the observed SOA formation. The authors should discuss the issue for details.

2. Adding AA and water (increasing relative humidity) should change the acidity of SOA. It is known that the pH dramatically influences the fates of ozonation products in condensed phase. See Zhao et al. J. Phys. Chem. A 2018, 122, 5190 and Qiu et al. Environ. Sci. Technol. 2020, doi.org/10.1021/acs.est.0c03438 for example. How does the change of SOA acidity influence the results?

3. It has been reported that water accelerates the decomposition of alpha-hydroxyalkyl-hydroperoxides (formed by SCIs + water) and alpha-acyloxy-hydroperoxides (formed by SCIs + carboxylic acids) [see Zhao et al. J. Phys. Chem. A 2018, 122, 5190, Qiu et al. Environ. Sci. Technol. 2020, 54, 3890−3899]. Could this humidity-assisted decomposition of ROOH explain the observed RH effects on SOA yield? The authors should comment on the issue in the text.

---

## Referee Comment (RC2) · Anonymous Referee #2 · 12 Sep 2020

This manuscript by Gong and Chen describes a series of laboratory experiments, aiming to elucidate the contribution of stabilized Criegee intermediates (SCIs) to the formation of SOA from limonene. The authors used a creative flow tube setup to investigate SCIs arising from ozonation of the endo- and exo-double bonds (DBs) separately. By employing an SCI scavenger in the system, the authors claimed that they have quantified the contribution of SCI chemistry towards the SOA yields as a function of RH. The major conclusion is that water plays a complex role in the reaction system, suppressing SOA formation under low RH, while facilitating SOA formation at high RH. Over the past few years, the importance of SCI chemistry in the atmosphere has become evident. With SCI being a reactive intermediate that is difficult to detect, quantitative evaluations for the importance of SCI is lacking. This manuscript aims to provide quantitative information that fills our gap in understanding. The topic is timely and is within the scope of ACP. The writing and data analyses were conducted with caution. However, I have concerns regarding a few approaches and assumptions that authors employ in the study. I recommend a major revision before publication on ACP.

**Major comments:**

1) I'm afraid that the contribution of SCI to the SOA formation may be exaggerated in the current setup due to the presence of a high concentration of butanol (OH scavenger). The authors provided an estimate that 12.5% of SCI has reacted with butanol, using a lower-band estimate for the SCI reactivity with alcohol. It is not convincing that the effect of butanol is "not important (Line 304)". Can the author perform an experiment with an aprotic OH scavenger (e.g., hexane) to experimentally confirm their assumption?

2) The fraction of SCI reacting with water ($SCI_l$) was estimated solely from the formation of $H_2O_2$.
   - How did the authors measure $H_2O_2$? Was it the gas phase $H_2O_2$ or particle phase?
   - Although I agree that $H_2O_2$ is the major decomposition product of α-hydroxyhydroperoxides (product of SCI + $H_2O$), the decomposition of α-hydroxyhydroperoxides is an equilibrium process and may not always proceed completely.
   - α-hydroxyhydroperoxides are not the only source of $H_2O_2$. It's known that $H_2O_2$ is generated in SOA extracts, likely due to the decomposition of larger organic peroxides.
   - In particular, Zhao et al. 2018, J. Phys. Chem. A reported $H_2O_2$ arising from the decomposition of hydroperoxyester, which is formed from SCI + organic acids. Although the mechanism is not completely clear, this implies that the product of SCI + AA may also give rise to $H_2O_2$.

Citation: Ran Zhao, Christopher M. Kenseth, Yuanlong Huang, Nathan F. Dalleska, Xiaobi M. Kuang, Jierou Chen, Suzanne E. Paulson, and John H. Seinfeld
The Journal of Physical Chemistry A 2018 122 (23), 5190-5201 DOI: 10.1021/acs.jpca.8b02195

3) The flow tube experiments employ tens of ppb of limonene with an excess amount of O3 for reactions. While these concentrations are rather typical for flow tube experiments, I think the author should discuss the feasibility of extrapolating their flow tube results to the real environment. As the authors point out, limonene mixing ratios are at the sub-ppb level for forest and urban environments. In my opinion, SCIs will predominantly react with water when organic concentrations are low. Thus, the SOA formation potential of SCIs they determine in the flow tube may or may not be applicable to the ambient conditions.

**Minor and Technical Comments**

- Line 155 "that" to "than"
- Line 307 the sentence: "The reactions of SCIs with the compounds formed from SCIs scavengers would not compensate the effect of the consumption of SCIs on SOA formation." is unclear. Please rephrase.
- I'm supportive of the authors' idea to provide an atmospheric implication of their findings by simulating three scenarios: forest, urban, and indoor. Instead of investigating all the RH for each scenario, I wonder if authors can constrain the RH to ranges that are more relevant to each scenario? For instance, the most comfortable RH range for human occupancy in an indoor environment is between 30 to 60%. It is unlikely we see an indoor that are extremely dry or wet.
- Acid anhydrides can be hydrolyzed to form organic acids. By any chance, can the acid anhydrides arising from SCI + AA be hydrolyzed at higher RH, regenerating AA?
- Line 391-392 - "On the other hand" is repeated in the same sentence.
- Line 173 - k(other) - I wonder if wall loss of SCIs should be included in k(other)?

---

## Author Comment (AC1) · 17 Nov 2020

Response to Reviewer #1

We gratefully thank you for your constructive comments and through review. Below are our point-by-point responses to your comments.

(Q=Question, A= Answer, C=Change in the revised manuscript)

General Comments: Gong and Chen report the experimental study on the formation of SOA during the ozonolysis of limonene, a class of important biogenic VOC in the atmosphere. They used flow tube reactors under different relative humidity (RH) to investigate the mechanism of SOA formation, especially the role of stabilized Criegee intermediates (SCIs). Their findings imply the different mechanisms of SOA formation at dry vs humid condition. The subject is within the scope of ACP and some findings seem important from the viewpoint of atmospheric aerosol chemistry. However, I am concerned about some critical issues that should be addressed.

A: We highly appreciate your comments and suggestions. The questions you mentioned are answered as follows.

Specific Comments:

Q1. I am most concerned about the side reaction of SCIs with 2- butanol. In page 10, the authors wrote "The rate constants of CH2OO reaction with methanol and ethanol were measured to be about $10^{-15}$ cm3 molecule−1 s−1 at 298 K (McGillen et al., 2017)", but this statement is incorrect. Actually, McGillen et al. (ACS Earth Space Chem. 2017, 1, 664−672) experimentally determined the rate constants k ∼ $10^{-13}$ cm3

molecule−1 s−1 for CH2OO + CH3OH/C2H5OH and k ∼ 4 x $10^{-14}$ cm3 molecule−1 s−1 for (CH3)2COO + CH3OH at ∼ 298 K. Furthermore, Tadayon et al. (J. Phys. Chem. A 2018, 122, 1, 258–268) reported the rate constants of (1.9 ± 0.5) × $10^{-13}$ cm3 molecule-1 s–1 for the reaction of CH2OO with 2-propanol at 295 K. Hence, the assumption that the rate constant of limonene-derived SCIs reaction with 2-butanol is $10^{-15}$ cm3 molecule−1 s−1 seems to be inadequate. If the authors assumed the rate constant to be $10^{-13}$ cm3 molecule−1 s−1, then the ratio of the amount of SCIs reacted with 2-butanol to the amount of SCIs reacted with AA would be much larger than the value authors claimed. Thus, SCIs in the presence of an excess amount of 2-butanol would be exclusively converted into alpha-alkoxyalkyl-hydroperoxides, that may contribute to the observed SOA formation. The authors should discuss the issue for details.

A1: Thanks for your suggestion and we regret that we did not calculate the impact of adding 2-butanol correctly. We have revaluated the effect of 2-butanol on SCIs reactions in the revised manuscript, and to better understand the structure-dependent reactivity of SCIs, the chemical structures of SCIs formed from endo-DB and exo-DB ozonolysis are provided in the Supplement. Besides, we also conducted experiments with cyclohexane and lower concentration of 2-butanol to confirm the impact of 2-butanol.

C1: Lines 307–331 in Sect. 3.3.2:

Previous studies reported that as for the reactions of SCIs with alcohols, the substitution group of alcohols had little effect on the reactions, while the structures of SCIs showed an obvious influence. The rate constants of $CH_2OO$ reaction with alcohols were about $10^{-13}$ $cm^3$ $molecule^{-1}$ $s^{-1}$ at 298 K (Tadayon et al., 2018), and it was slower for $(CH_3)_2COO$ reaction with alcohols, whose rate constants were reported to be $10^{-15}$ ~ $10^{-14}$ $cm^3$ $molecule^{-1}$ $s^{-1}$ (Aroeira et al., 2019; McGillen et al., 2017). Watson et al. (2019) computed that the rate constant of *syn*-$CH_3CHOO$ + $CH_3OH$ was about $10^{-17}$ $cm^3$ $molecule^{-1}$ $s^{-1}$, which was much smaller than that of about $10^{-12}$ $cm^3$ $molecule^{-1}$ $s^{-1}$ for *anti*-$CH_3CHOO$ + $CH_3OH$ at 298 K. In limonene ozonolysis, several kinds of SCIs were produced and their structures were shown in Fig. S7. A mono-substituted SCIs and a di-substituted SCIs were formed from endo-DB ozonolysis, and exo-DB ozonolysis produced $CH_2OO$ and a di-substituted SCIs. When assuming the rate constant of SCIs reaction with 2-butanol as $10^{-14}$ $cm^3$ $molecule^{-1}$ $s^{-1}$, it was estimated that in terms of the concentrations of AA used in experiments, the ratio of the amount of SCIs reacted with 2-butanol to the amount of SCIs reacted with AA ranged from 0.06 to 1.25, and this ratio might be higher in exo-DB oxidation because of the formation of $CH_2OO$. This meant that part of SCIs could react with 2-butanol, producing α-alkoxyalkyl-hydroperoxides and contributing to the observed SOA, especially when using low concentrations of AA and water. To figure out whether the SOA formation potentials of SCIs estimated here were higher than those under the situation without 2-butanol, the experiments with cyclohexane as OH scavenger were carried out, and the details were described in the Supplement. It was found that with the use of cyclohexane, SCIs reactions still accounted for more than 60% in SOA formation and according to the fitting results, the SOA formation potentials of SCIs were even a bit larger than those with the use of 2-butanol, and their deviations were within 12%. This phenomenon was speculated to be due to the higher concentration of $RO_2$ radicals when using cyclohexane, promoting the reactions of SCIs with $RO_2$. To further determine the impact of 2-butanol, we also conducted experiments with the concentration of 2-butanol as about 150 ppmv, which was half of the previous concentration of 2-butanol used in experiments and was estimated to be sufficient for scavenging more than 99% OH radicals. It was observed that the amount of SOA formation was not impacted by the concentration of 2-butanol, and the SOA formation potentials of SCIs under a lower concentration of 2-butanol were similar with those under a higher concentration of 2-butanol. Based on the results elaborated above, we confirmed that the effect of 2-butanol on the results was limited.

Q2. Adding AA and water (increasing relative humidity) should change the acidity of SOA. It is known that the pH dramatically influences the fates of ozonation products in condensed phase. See Zhao et al. J. Phys. Chem. A 2018, 122, 5190 and Qiu et al. Environ. Sci. Technol. 2020, doi.org/10.1021/acs.est.0c03438 for example. How does the change of SOA acidity influence the results?

A2: Thanks for your suggestion. The acidity of aerosols is an issue needing consideration and we have added a discussion on the effect of SOA acidity on the results in the revised manuscript.

C2: Lines 344–361 in Sect. 3.3.3:

In this study, with the addition of AA the acidity of aerosols would change, and some reactions that happened in bulk phase were influenced, especially under high-humidity conditions. In view of this, the effect of SOA acidity on particle-phase reactions was considered. Zhao et al. (2018) investigated the aqueous-phase hydrolysis of α-acyloxyalkyl-hydroperoxides, which were produced from reactions of SCIs and organic acids. It was found that α-acyloxyalkyl-hydroperoxides decomposed promptly when pH was larger than 5 in aqueous phase. The aqueous decomposition of α-hydroxyalkyl-hydroperoxides was also reported to be accelerated by acids, and the rate coefficients of decay increased with decreasing pH (Qiu et al., 2020a). These studies implied that acids, acting as catalysts, promoted the decomposition processes of some compounds in liquid particles and might reduce the amount of SOA. However, the effect of acidity on bulk-phase reactions was found to be complicated, and Iinuma et al. (2004) reported that acidity promoted the formation of large molecules in particles. Some reactions, which produced peroxyacetals, esters, aldols, etc., and contributed to SOA formation, could be catalyzed by acids and $H^+$ in aqueous phase (Ziemann and Atkinson, 2012), while the rate coefficients of these reactions with varying pH were not clear. Actually, we could not evaluate the effect of acidity on the formation of SOA accurately because the acidity of aerosols might impact a series of reactions, and the mechanisms and rates of these reactions at different pH were vague. It was noted that Chen et al. (2008) found that in the aqueous-phase ozonolysis of methacrolein and methyl vinyl ketone, the yields of products were almost independent of pH, and Zhang et al. (2009) also observed this phenomenon in the ozonolysis of α-pinene and β-pinene in aqueous phase. Thus the effect of the particle acidity on the results might also be limited here, and the accurate estimates of this issue still needed further research.

Q3. It has been reported that water accelerates the decomposition of alpha-hydroxyalkyl-hydroperoxides (formed by SCIs + water) and alpha-acyloxy-hydroperoxides (formed by SCIs + carboxylic acids) [see Zhao et al. J. Phys. Chem. A 2018, 122, 5190, Qiu et al. Environ. Sci. Technol. 2020, 54, 3890−3899]. Could this humidity-assisted decomposition of ROOH explain the observed RH effects on SOA yield? The authors should comment on the issue in the text.

A3: Thanks for your suggestion. Water could accelerate the decomposition processes of α-acyloxyalkyl-hydroperoxides and α-hydroxyalkyl-hydroperoxides in aqueous phase, resulting in the formation of $H_2O_2$. The reactions of $H_2O_2$ in particles have been thought to be important and may impact the aerosols formation. A discussion about this issue was provided in the revised manuscript.

C3: Lines 368–377 in Sect. 3.3.3:

On the other hand, a significant amount of $H_2O_2$ formed from SCIs reaction with water was observed in ozonolysis under high-humidity conditions (Chen et al., 2016; Jiang et al., 2013), and water was reported to accelerate the decomposition of α-acyloxyalkyl-hydroperoxides and α-hydroxyalkyl-hydroperoxides in aqueous phase, resulting in the formation of $H_2O_2$ (Qiu et al., 2019, 2020b; Zhao et al., 2018). $H_2O_2$ was reported to play an important role in the nonradical oxidation of carbonyls in aqueous phase (Galloway et al., 2011; Herrmann et al., 2015), producing hydroxyhydroperoxides and promoting SOA formation (Zhao et al., 2012, 2013). The $H_2O_2$ oxidation of carbonyls was speculated to mainly occur in surface liquid layer of aerosols, resulting in the generation of organic peroxides and high-molecular-weight oligomers (Sui et al., 2017; Zhang et al., 2019). The impact of $H_2O_2$ reactions at air-liquid interface might be another reason for the performances of SCIs in SOA formation under high-humidity conditions.

**Figure S7.** Structures of limonene-derived SCIs formed from endo-DB and exo-DB ozonolysis.

**References**

[revised manuscript text omitted]

---

## Author Comment (AC2) · 17 Nov 2020

Response to Reviewer #2

We gratefully thank you for your constructive comments and through review. Below are our point-by-point responses to your comments.

(Q=Question, A= Answer, C=Change in the revised manuscript)

General Comments: This manuscript by Gong and Chen describes a series of laboratory experiments, aiming to elucidate the contribution of stabilized Criegee intermediates (SCIs) to the formation of SOA from limonene. The authors used a creative flow tube setup to investigate SCIs arising from ozonation of the endo- and exo-double bonds (DBs) separately. By employing an SCI scavenger in the system, the authors claimed that they have quantified the contribution of SCI chemistry towards the SOA yields as a function of RH. The major conclusion is that water plays a complex role in the reaction system, suppressing SOA formation under low RH, while facilitating SOA formation at high RH. Over the past few years, the importance of SCI chemistry in the atmosphere has become evident. With SCI being a reactive intermediate that is difficult to detect, quantitative evaluations for the importance of SCI is lacking. This manuscript aims to provide quantitative information that fills our gap in understanding. The topic is timely and is within the scope of ACP. The writing and data analyses were conducted with caution. However, I have concerns regarding a few approaches and assumptions that authors employ in the study. I recommend a major revision before publication on ACP.

A: We highly appreciate your comments and suggestions. The questions you mentioned are answered as follows.

**Major comments:**

Q1. I'm afraid that the contribution of SCI to the SOA formation may be exaggerated in the current setup due to the presence of a high concentration of butanol (OH scavenger). The authors provided an estimate that 12.5% of SCI has reacted with butanol, using a lower-band estimate for the SCI reactivity with alcohol. It is not convincing that the effect of butanol is "not important (Line 304)". Can the author perform an experiment with an aprotic OH scavenger (e.g., hexane) to experimentally confirm their assumption?

A1: Thanks for your suggestion. Here to evaluate whether the addition of 2-butanol made obvious impacts on the results, we conducted experiments with cyclohexane as OH scavenger, and the results were discussed as follows.

Part of SCIs might react with 2-butanol, producing α-alkoxyalkyl-hydroperoxides and contributing to the observed SOA, especially when using low concentrations of AA and

water. To figure out whether the SOA formation potentials of SCIs estimated here were higher than those under the situation without 2-butanol, the experiments with cyclohexane as OH scavenger were carried out. Here three representative conditions: dry conditions, 40% RH (representing low-humidity conditions) and 80% RH (representing high-humidity conditions), were analyzed in the endo-DB ozonolysis. The abilities of 2-butanol and cyclohexane on scavenging OH radicals were similar (Chew and Atkinson, 1996), however, the use of different OH scavengers brought different impacts on the reaction system. When 2-butanol was used, higher $[HO_2]/[RO_2]$ was observed, which was thought to be more similar to the atmospheric conditions, while adding cyclohexane resulted in a lower $[HO_2]/[RO_2]$ (Docherty and Ziemann, 2003; Jonsson et al., 2008). In view of this, this study chose 2-butanol as OH scavenger, while the use of cyclohexane could provide a contrast to help us better understand the mechanisms in the reaction system.

In the experiments with cyclohexane, the concentrations of limonene and $O_3$ were about 90 ppbv and 270 ppbv, and the reaction time was 240 s. Around 400 ppmv of cyclohexane was added to scavenge OH radicals. With the addition of cyclohexane, the SOA yields were found to be lower than those with the addition of 2-butanol, suggesting that higher concentration of $HO_2$ radicals promoted aerosols formation (Keywood et al., 2004). Through adding different concentrations of AA (24–480 ppbv), the amount of SCIs in the reaction system was regulated and calculated as elaborated in Sect. 3.1, and the dependence of SOA mass concentration on the amount of SCIs was shown in Fig. S8. The SCIs reactions still accounted for more than 60% in SOA formation and according to the fitting results, the SOA formation potentials of SCIs under the use of cyclohexane were even a bit larger than those under the use of 2-butanol, and the deviations were within 12%. This phenomenon was speculated to be due to the higher concentration of $RO_2$ radicals when using cyclohexane, promoting the reactions of SCIs with $RO_2$.

To further determine the impact of 2-butanol, we also conducted experiments with the concentration of 2-butanol as about 150 ppmv, which was half of the previous concentration of 2-butanol used in experiments and was estimated to be sufficient for scavenging more than 99% OH radicals. It was observed that the amount of SOA formation was not impacted by the concentration of 2-butanol, and the SOA formation potentials of SCIs under a lower concentration of 2-butanol were similar with those under a higher concentration of 2-butanol. Based on the results elaborated above, we confirmed that the effect of 2-butanol on the results was limited. We have added the details on the experiments with cyclohexane in the Supplement.

Q2. The fraction of SCI reacting with water ($SCI_I$) was estimated solely from the

formation of $H_2O_2$.

- How did the authors measure $H_2O_2$? Was it the gas phase $H_2O_2$ or particle phase?

75 - Although I agree that $H_2O_2$ is the major decomposition product of α-hydroxyhydroperoxides (product of SCI + $H_2O$), the decomposition of α-hydroxyhydroperoxides is an equilibrium process and may not always proceed completely.

- α-hydroxyhydroperoxides are not the only source of $H_2O_2$. It's known that $H_2O_2$ is
80 generated in SOA extracts, likely due to the decomposition of larger organic peroxides.

- In particular, Zhao et al. 2018, J. Phys. Chem. A reported $H_2O_2$ arising from the decomposition of hydroperoxyester, which is formed from SCI + organic acids. Although the mechanism is not completely clear, this implies that the product of SCI + AA may also give rise to $H_2O_2$.

85 Citation: Ran Zhao, Christopher M. Kenseth, Yuanlong Huang, Nathan F. Dalleska, Xiaobi M. Kuang, Jierou Chen, Suzanne E. Paulson, and John H. Seinfeld, The Journal of Physical Chemistry A 2018 122 (23), 5190-5201 DOI: 10.1021/acs.jpca.8b02195

A2: Thanks for your suggestions. When we estimated the yield of $SCI_I$ in limonene ozonolysis, the formation of gas-phase $H_2O_2$ was measured, which was elaborated in
90 our previous study (Gong et al., 2018). For the detection of gas-phase $H_2O_2$, the gas samples passing through the PTFE filter were collected in a glass coil collector at a temperature of 4 °C with $H_3PO_4$ solution (pH 3.5) serving as the rinsing solution. After the collection, solutions containing peroxides were analyzed by HPLC (Agilent 1100, USA) coupled with post-column derivatization and fluorescence detection online.
95 Peroxides separated by column chromatography reacted with *p*-hydroxyphenylacetic acid (POPHA) to form POPHA dimers under the catalysis of hemin, and then the dimers were quantified using a fluorescence detector. With the increase of RH, it was observed that the yield of $H_2O_2$ increased significantly from dry conditions to 70% RH, and the $H_2O_2$ yield approached the limiting value above 70% RH, suggesting that reaction with
100 water suppressed other reactions of $SCI_I$. In the exo-DB oxidation, the formation of hydroxymethyl hydroperoxide was also taken into consideration. Through the box model simulation, the contribution of $HO_2$ self-reaction to $H_2O_2$ formation was estimated to be limited. As for the reaction of SCIs with water, the products α-hydroxyalkyl hydroperoxides were reported to be preferential to decompose and
105 generate $H_2O_2$ (Chen et al., 2016; Kumar et al., 2014). Although theoretical calculations indicated that the decomposition of α-hydroxyalkyl hydroperoxides was slow, some studies proved that water and acids could significantly catalyze the process (Anglada et al., 2002, 2011; Aplincourt and Anglada, 2003; Crehuet et al., 2001), and $H_2O_2$

formation occurred rapidly (Chen et al., 2016; Winterhalter et al., 2000). In addition, few α-hydroxyalkyl hydroperoxides larger than hydroxymethyl hydroperoxide were identified in gas phase, and the decomposition of α-hydroxyalkyl hydroperoxides was speculated to be totally completed.

The generation of $H_2O_2$ from aerosols in aqueous phase, which is mainly due to the decomposition of some compounds, has received attentions in recent years (Wang et al., 2011; Zhao et al., 2018). Here only the gas-phase $H_2O_2$ was detected to estimate the yield of $SCI_I$, while it was still needed to analyze whether the formation of $H_2O_2$ in SOA could impact the results. Zhao et al. (2018) reported the aqueous decomposition rate coefficients of α-acyloxyalkyl hydroperoxides, whose lifetime was estimated to be about 24 min in liquid aerosols. As the reaction time in flow tube reactors was around 4 min, the $H_2O_2$ formation from aerosols was not considered to contribute much to gas-phase $H_2O_2$ in this study. In addition, it was found that the dependence of $H_2O_2$ yield on RH could be well simulated with the gas-phase mechanisms, confirming that the particle-phase formation of $H_2O_2$ did not make obvious impact on the results. We have added the discussion about the formation and measurement of $H_2O_2$ in the Supplement.

Q3. The flow tube experiments employ tens of ppb of limonene with an excess amount of $O_3$ for reactions. While these concentrations are rather typical for flow tube experiments, I think the author should discuss the feasibility of extrapolating their flow tube results to the real environment. As the authors point out, limonene mixing ratios are at the sub-ppb level for forest and urban environments. In my opinion, SCIs will predominantly react with water when organic concentrations are low. Thus, the SOA formation potential of SCIs they determine in the flow tube may or may not be applicable to the ambient conditions.

A3: Thanks for your suggestion and a discussion on the effect of the concentrations of reactants has been added in the Supplement. In this study, to get enough products for analysis in a short reaction time, both of the concentrations of limonene and $O_3$ used in experiments were higher than those in the real atmosphere, and it was needed to consider the effect of concentrations of reactants. In the atmosphere, the concentrations of organic compounds formed from limonene ozonolysis are much smaller than those in flow tube reactors, while it should be noted that limonene-derived SCIs would not only react with the compounds formed from limonene, they could also react with other compounds in the ambient air. In this study, we determined the rate of SCIs isomerization and reaction with other products. In the atmosphere, the organic compounds that SCIs could react with are generally carboxylic acids, carbonyls, alcohols, and $RO_2$ radicals, and the concentrations of these compounds in forest are about $10^{11}$ molecule $cm^{-3}$, $10^{11}$ molecule $cm^{-3}$, $10^{11}$ molecule $cm^{-3}$, and $10^9$ molecule

cm$^{-3}$, respectively. In urban area, the concentrations of carbonyls and alcohols were reported to be higher because of the anthropogenic emissions (Vereecken et al., 2012). The rate coefficients of SCIs reaction with carboxylic acids, carbonyls, alcohols, and RO$_2$ radicals were reported to be around $10^{-10}$ molecule cm$^3$ s$^{-1}$, $10^{-12}$ molecule cm$^3$ s$^{-1}$, $10^{-14}$ molecule cm$^3$ s$^{-1}$, and $10^{-11}$ molecule cm$^3$ s$^{-1}$, respectively (Khan et al., 2018; Lin and Chao, 2017; Zhao et al., 2017). It was estimated that the sum of the rate of SCIs isomerization and reaction with organic compounds in the atmosphere was on the same order of magnitude as that in experiments, and thus the results obtained here were considered to be feasible to the ambient conditions. We declare that further studies on different concentrations of reactants with the coexistence of other organic compounds would make the results more concise.

**Minor and Technical Comments:**

Q4. Line 155 "that" to "than"

A4: We have revised that.

Q5. Line 307 the sentence: "The reactions of SCIs with the compounds formed from SCIs scavengers would not compensate the effect of the consumption of SCIs on SOA formation." is unclear. Please rephrase.

A5: Thanks for your suggestion. We have revised this sentence as follows.

C5: Lines 335–336 in Sect. 3.3.2:

The major role of SCIs scavengers was consuming SCIs in the reaction system, and the effect of products formed from SCIs scavengers on SCIs reactions was not expected to be important.

Q6. I'm supportive of the authors' idea to provide an atmospheric implication of their findings by simulating three scenarios: forest, urban, and indoor. Instead of investigating all the RH for each scenario, I wonder if authors can constrain the RH to ranges that are more relevant to each scenario? For instance, the most comfortable RH range for human occupancy in an indoor environment is between 30 to 60%. It is unlikely we see an indoor that are extremely dry or wet.

A6: Thanks for your suggestion. The relevant RH under each scenario was considered and provided in the revised manuscript.

C6: Lines 413–418 in Sect. 3.4:

In forest, the typical RH is higher than 60%, and for urban and indoor area, they are usually under low-humidity conditions (Carslaw, 2017; Vereecken et al., 2012). According to the SOA formation potential of SCIs, it is estimated that the typical

180 contribution of limonene-derived SCIs to SOA formation is $(8.21 \pm 0.15) \times 10^{-2}$ μg m$^{-3}$ h$^{-1}$ in forest, $(4.26 \pm 0.46) \times 10^{-2}$ μg m$^{-3}$ h$^{-1}$ in urban area, and $(2.52 \pm 0.28) \times 10^{-1}$ μg m$^{-3}$ h$^{-1}$ in indoor area.

Q7. Acid anhydrides can be hydrolyzed to form organic acids. By any chance, can the acid anhydrides arising from SCI + AA be hydrolyzed at higher RH, regenerating AA?

185 A7: Thanks for your suggestion. Some acid anhydrides were stable against water (Rong et al., 2020), and some others could be hydrolyzed to form organic acids in aqueous phase (Fritzler et al., 2014). The hydrolysis reactions of acid anhydrides were usually accelerated significantly by some catalysts (Faria et al., 2008). In the gas phase, α-acyloxyalkyl hydroperoxides formed from SCIs reaction with AA could dehydrate and

190 form acid anhydrides (Long et al., 2009), and the reintroduction of water on anhydrides might not be easy to happen. The hydrolysis of acid anhydrides was expected to occur after anhydrides were taken up into atmospheric liquid water (Taatjes et al., 2019). Considering that the reaction time in this study was just a few minutes, this process was not predicted to be important.

195 Q8. Line 391-392 - "On the other hand" is repeated in the same sentence.

A8: Thanks for your suggestion. We have revised that in the manuscript.

Q9. Line 173 - k(other) - I wonder if wall loss of SCIs should be included in k(other)?

A9: Yes. Although the wall loss of SCIs was difficult to estimate and was not discussed in detail, we think the rate of wall loss of SCIs could be included in $k_{(other)}$.

[Figure]

200

**Figure S8.** The dependence of SOA mass concentration on the amount of SCIs at different relative humidity (RH) in the first-generation oxidation with cyclohexane as OH scavenger.

205 **Reference**

Anglada, J. M., Aplincourt, P., Bofill, J. M., and Cremer, D.: Atmospheric formation of OH radicals and $H_2O_2$ from alkene ozonolysis under humid conditions, Chem. Phys. Chem., 3, 215–221, doi: 10.1002/1439-7641(20020215)3:2<215::Aid-Cphc215>3.3.Co;2-V, 2002.

210 Anglada, J. M., González, J., and Torrent-Sucarrat, M.: Effects of the substituents on the reactivity of carbonyl oxides. A theoretical study on the reaction of substituted carbonyl oxides with water, Phys. Chem. Chem. Phys., 13, 13034–13045, doi: 10.1039/C1CP20872A, 2011.

Aplincourt, P. and Anglada, J. M.: Theoretical studies of the isoprene ozonolysis under 215 tropospheric conditions. 2. Unimolecular and water-assisted decomposition of the α-hydroxy hydroperoxides, J. Phys. Chem. A, 107, 5812–5820, doi: 10.1021/jp034203w, 2003.

Carslaw, N.: A new detailed chemical model for indoor air pollution, Atmos. Environ., 41, 1164–1179, doi: 10.1016/j.atmosenv.2006.09.038, 2007.

220 Chen, L., Wang, W. L., Wang, W. N., Liu, Y. L., Liu, F. Y., Liu, N., and Wang, B. Z.: Water-catalyzed decomposition of the simplest Criegee intermediate $CH_2OO$, Theor. Chem. Acc., 135, 131, doi: 10.1007/s00214-016-1894-9, 2016.

Chew, A. A. and Atkinson, R.: OH radical formation yields from the gas-phase reactions of $O_3$ with alkenes and monoterpenes, J. Geophys. Res., 101, 28649–28653, doi: 225 10.1029/96JD02722, 1996.

Crehuet, R., Anglada, J. M., and Bofill, J. M.: Tropospheric formation of hydroxymethyl hydroperoxide, formic acid, $H_2O_2$, and OH from carbonyl oxide in the presence of water vapor: A theoretical study of the reaction mechanism, Chem. Eur. J., 7, 2227–2235, doi: 10.1002/1521-3765(20010518)7:10<2227::AID-230 CHEM2227>3.0.CO;2-O, 2001.

Docherty, K. S. and Ziemann, P. J.: Effects of stabilized Criegee intermediate and OH radical scavengers on aerosol formation from reactions of β-pinene with $O_3$, Aerosol Sci. Technol., 37, 877–891, doi: 10.1080/02786820300930, 2003.

Faria, A. C., Mello, R. S., Orth, E. S., and Nome, F.: Hydrolysis of benzoic anhydride 235 mediated by ionenes and micelles, J. Mol. Catal. A: Chem., 289, 106−111, doi: 10.1016/j.molcata.2008.04.019, 2008.

Fritzler, B. C., Dharmavaram, S., Hartrim, R. T., and Diffendall, G. F.: Acetic anhydride hydrolysis at high acetic anhydride to water ratios, Int. J. Chem. Kinet., 46, 151−160,

doi: 10.1002/kin.20838, 2014.

Gong, Y. W., Chen, Z. M., and Li, H.: The oxidation regime and SOA composition in limonene ozonolysis: roles of different double bonds, radicals, and water, Atmos. Chem. Phys., 18, 15105–15123, doi: 10.5194/acp-18-15105-2018, 2018.

Jonsson, A. M., Hallquist, M., and Ljungström, E.: Influence of OH scavenger on the water effect on secondary organic aerosol formation from ozonolysis of limonene, 3-carene, and α-pinene, Environ. Sci. Technol., 42, 5938–5944, doi: 10.1021/es702508y, 2008.

Keywood, M. D., Kroll, J. H., Varutbangkul, V., Bahreini, R., Flagan, R. C., and Seinfeld, J. H.: Secondary organic aerosol formation from cyclohexene ozonolysis: Effect of OH scavenger and the role of radical chemistry, Environ. Sci. Technol., 38, 3343–3350, doi: 10.1021/es049725j, 2004.

Khan, M. A. H., Percival, C. J., Caravan, R. L., Taatjes, C. A., and Shallcross, D. E.: Criegee intermediates and their impacts on the troposphere, Environ. Sci.: Processes Impacts, 20, 437–453, doi: 10.1039/C7EM00585G, 2018.

Kumar, M., Busch, D. H., Subramaniam, B., and Thompson, W. H.: Role of tunable acid catalysis in decomposition of hydroxyalkyl hydroperoxides and mechanistic implications for tropospheric chemistry, J. Phys. Chem. A, 118, 9701–9711, doi: 10.1021/jp505100x, 2014.

Lin, J. J. M. and Chao, W.: Structure-dependent reactivity of Criegee intermediates studied with spectroscopic methods, Chem. Soc. Rev., 46, 7483–7497, doi: 10.1039/C7CS00336F, 2017.

Long, B., Cheng, J. R., Tan, X. F., and Zhang, W. J.: Theoretical study on the detailed reaction mechanisms of carbonyl oxide with formic acid, J. Mol. Struct.: THEOCHEM, 916, 159−167, doi: 10.1016/j.theochem.2009.09.028, 2009.

Rong, H., Liu, L., Liu, J. R., and Zhang, X. H.: Glyoxylic sulfuric anhydride from the gas-phase reaction between glyoxylic acid and $SO_3$: a potential nucleation precursor, J. Phys. Chem. A, 124, 3261−3268, doi: 10.1021/acs.jpca.0c01558, 2020.

Taatjes, C. A., Khan, M. A. H., Eskola, A. J., Percival, C. J., Osborn, D. L., Wallington, T. J., and Shallcross, D. E.: Reaction of perfluorooctanoic acid with Criegee intermediates and implications for the atmospheric fate of perfluorocarboxylic acids, Environ. Sci. Technol., 53, 1245−1251, doi: 10.1021/acs.est.8b05073, 2019.

Vereecken, L., Harder, H., and Novelli, A.: The reaction of Criegee intermediates with NO, $RO_2$, and $SO_2$, and their fate in the atmosphere, Phys. Chem. Chem. Phys., 14,

14682–14695, doi: 10.1039/c2cp42300f, 2012.

Wang, Y., Kim, H., and Paulson, S. E.: Hydrogen peroxide generation from α- and β-pinene and toluene secondary organic aerosols, Atmos. Environ., 45, 3149–3156, doi: 10.1016/j.atmosenv.2011.02.060, 2011.

Winterhalter, R., Neeb, P., Grossmann, D., Kolloff, A., Horie, O., and Moortgat, G.: Products and mechanism of the gas phase reaction of ozone with β-pinene, J. Atmos. Chem, 35, 165–197, doi: 10.1023/A:1006257800929, 2000.

Zhao, Q. L., Liu, F. Y., Wang, W. N., Li, C. Y., Lü, J., and Wang, W. L.: Reactions between hydroxyl-substituted alkylperoxy radicals and Criegee intermediates: correlations of the electronic characteristics of methyl substituents and the reactivity, Chem. Chem. Phys., 19, 15073, doi: 10.1039/c7cp00869d, 2017.

Zhao, R., Kenseth, C. M., Huang, Y. L., Dalleska, N. F., Kuang, X. M., Chen, J. R., Paulson, S. E., and Seinfeld, J. H.: Rapid aqueous-phase hydrolysis of ester hydroperoxides arising from Criegee intermediates and organic acids, J. Phys. Chem. A, 122, 5190−5201, doi: 10.1021/acs.jpca.8b02195, 2018.

---

## Author Response (AR1)

November 16, 2020

ACP Editor

Dear Prof. Nga Lee Ng,

Enclosed please find our revised manuscript entitled "Quantification of the role of

5 *stabilized Criegee intermediates in the formation of aerosols in limonene ozonolysis*", revised supplement and two responses to the anonymous referees #1 and #2. We gratefully appreciate the reviewers for their constructive suggestions to help us improve the manuscript. We revised the manuscript by responding to all comments point by point. We sincerely hope the revised manuscript is suitable for publication on ACP.

**10 The major revisions are specified as follows:**

1. The effect of 2-butanol on SCIs reactions was revaluated in the revised manuscript, and the experiments with cyclohexane as OH scavenger were conducted.

2. Discussion on the effect of aerosols acidity on the results was added in the manuscript.

3. Details on the formation and measurement of H2O2 were added in the Supplement.

15 4. Discussion on the effect of the reactants concentrations on the results has been added in the Supplement.

Detailed changes made in the manuscript can be seen in the marked-up version in this response.

Thanks for your time.

20 Sincerely yours,

Zhongming Chen and Yiwei Gong

**Response to Reviewer #1**

25 We gratefully thank you for your constructive comments and through review. Below are our point-by-point responses to your comments.

**(Q=Question, A= Answer, C=Change in the revised manuscript)**

General Comments: Gong and Chen report the experimental study on the formation of SOA during the ozonolysis of limonene, a class of important biogenic VOC in the atmosphere. They used flow tube reactors under different relative humidity (RH) to

30 atmosphere. They used flow tube reactors under different relative humidity (RH) to investigate the mechanism of SOA formation, especially the role of stabilized Criegee intermediates (SCIs). Their findings imply the different mechanisms of SOA formation at dry vs humid condition. The subject is within the scope of ACP and some findings seem important from the viewpoint of atmospheric aerosol chemistry. However, I am 35 concerned about some critical issues that should be addressed.

A: We highly appreciate your comments and suggestions. The questions you mentioned are answered as follows.

**Specific Comments:**

Q1. I am most concerned about the side reaction of SCIs with 2- butanol. In page 10,
the authors wrote "The rate constants of CH2OO reaction with methanol and ethanol were measured to be about 10-15 cm3 molecule-1 s-1 at 298 K (McGillen et al., 2017)", but this statement is incorrect. Actually, McGillen et al. (ACS Earth Space Chem. 2017, 1, 664–672) experimentally determined the rate constants k ~ 10-13 cm3 molecule-1 s-1 for CH2OO + CH3OH/C2H5OH and k ~ 4 x 10-14 cm3 molecule-1
s-1 for (CH3)2COO + CH3OH at ~ 298 K. Furthermore, Tadayon et al. (J. Phys. Chem. A 2018, 122, 1, 258–268) reported the rate constants of (1.9 ± 0.5) × 10-13 cm3 molecule-1 s-1 for the reaction of CH2OO with 2-propanol at 295 K. Hence, the assumption that the rate constant of limonene-derived SCIs reaction with 2-butanol is 10-15 cm3 molecule-1 s-1 seems to be inadequate. If the authors assumed the rate constant to be 10-13 cm3 molecule-1 s-1, then the ratio of the amount of SCIs reacted

with 2-butanol to the amount of SCIs reacted with AA would be much larger than the value authors claimed. Thus, SCIs in the presence of an excess amount of 2-butanol would be exclusively converted into alpha-alkoxyalkyl-hydroperoxides, that may contribute to the observed SOA formation. The authors should discuss the issue for details.

A1: Thanks for your suggestion and we regret that we did not calculate the impact of adding 2-butanol correctly. We have revaluated the effect of 2-butanol on SCIs reactions in the revised manuscript, and to better understand the structure-dependent

reactivity of SCIs, the chemical structures of SCIs formed from endo-DB and exo-DB

60 ozonolysis are provided in the Supplement. Besides, we also conducted experiments with cyclohexane and lower concentration of 2-butanol to confirm the impact of 2-butanol.

C1: Lines 307–331 in Sect. 3.3.2:

- Previous studies reported that as for the reactions of SCIs with alcohols, the substitution
  group of alcohols had little effect on the reactions, while the structures of SCIs showed an obvious influence. The rate constants of CH2OO reaction with alcohols were about 10-13 cm3 molecule-1 s-1 at 298 K (Tadayon et al., 2018), and it was slower for (CH3)2COO reaction with alcohols, whose rate constants were reported to be 10-15 ~ 10-14 cm3 molecule-1 s-1 (Aroeira et al., 2019; McGillen et al., 2017). Watson et al.
  (2019) computed that the rate constant of *syn*-CH3CHOO + CH3OH was about 10-17 cm3 molecule-1 s-1, which was much smaller than that of about 10-12 cm3 molecule-1 s-1 for *anti*-CH3CHOO + CH3OH at 298 K. In limonene ozonolysis, several kinds of
  - SCIs were produced and their structures were shown in Fig. S7. A mono-substituted SCIs and a di-substituted SCIs were formed from endo-DB ozonolysis, and exo-DB
- 75 ozonolysis produced CH2OO and a di-substituted SCIs. When assuming the rate constant of SCIs reaction with 2-butanol as  $10^{-14}$  cm3 molecule-1 s-1, it was estimated that in terms of the concentrations of AA used in experiments, the ratio of the amount of SCIs reacted with 2-butanol to the amount of SCIs reacted with AA ranged from 0.06 to 1.25, and this ratio might be higher in exo-DB oxidation because of the
- 80 formation of CH2OO. This meant that part of SCIs could react with 2-butanol, producing  $\alpha$ -alkoxyalkyl-hydroperoxides and contributing to the observed SOA, especially when using low concentrations of AA and water. To figure out whether the SOA formation potentials of SCIs estimated here were higher than those under the situation without 2-butanol, the experiments with cyclohexane as OH scavenger were
- 85 carried out, and the details were described in the Supplement. It was found that with the use of cyclohexane, SCIs reactions still accounted for more than 60% in SOA formation and according to the fitting results, the SOA formation potentials of SCIs were even a bit larger than those with the use of 2-butanol, and their deviations were within 12%. This phenomenon was speculated to be due to the higher concentration of
- 90 RO2 radicals when using cyclohexane, promoting the reactions of SCIs with RO2. To further determine the impact of 2-butanol, we also conducted experiments with the concentration of 2-butanol as about 150 ppmv, which was half of the previous concentration of 2-butanol used in experiments and was estimated to be sufficient for scavenging more than 99% OH radicals. It was observed that the amount of SOA
- 95 formation was not impacted by the concentration of 2-butanol, and the SOA formation potentials of SCIs under a lower concentration of 2-butanol were similar with those

under a higher concentration of 2-butanol. Based on the results elaborated above, we confirmed that the effect of 2-butanol on the results was limited.

Q2. Adding AA and water (increasing relative humidity) should change the acidity of
SOA. It is known that the pH dramatically influences the fates of ozonation products in
condensed phase. See Zhao et al. J. Phys. Chem. A 2018, 122, 5190 and Qiu et al.
Environ. Sci. Technol. 2020, doi.org/10.1021/acs.est.0c03438 for example. How does
the change of SOA acidity influence the results?

A2: Thanks for your suggestion. The acidity of aerosols is an issue needing
consideration and we have added a discussion on the effect of SOA acidity on the results in the revised manuscript.

C2: Lines 344–361 in Sect. 3.3.3:

110

In this study, with the addition of AA the acidity of aerosols would change, and some reactions that happened in bulk phase were influenced, especially under high-humidity conditions. In view of this, the effect of SOA acidity on particle-phase reactions was considered. Zhao et al. (2018) investigated the aqueous-phase hydrolysis of  $\alpha$ -acyloxyalkyl-hydroperoxides, which were produced from reactions of SCIs and organic

acids. It was found that  $\alpha$ -acyloxyalkyl-hydroperoxides decomposed promptly when

- pH was larger than 5 in aqueous phase. The aqueous decomposition of α-hydroxyalkylhydroperoxides was also reported to be accelerated by acids, and the rate coefficients of decay increased with decreasing pH (Qiu et al., 2020a). These studies implied that acids, acting as catalysts, promoted the decomposition processes of some compounds in liquid particles and might reduce the amount of SOA. However, the effect of acidity on bulk-phase reactions was found to be complicated, and linuma et al. (2004) reported
- 120 that acidity promoted the formation of large molecules in particles. Some reactions, which produced peroxyacetals, esters, aldols, etc., and contributed to SOA formation, could be catalyzed by acids and H+ in aqueous phase (Ziemann and Atkinson, 2012), while the rate coefficients of these reactions with varying pH were not clear. Actually, we could not evaluate the effect of acidity on the formation of SOA accurately because
- 125 the acidity of aerosols might impact a series of reactions, and the mechanisms and rates of these reactions at different pH were vague. It was noted that Chen et al. (2008) found that in the aqueous-phase ozonolysis of methacrolein and methyl vinyl ketone, the yields of products were almost independent of pH, and Zhang et al. (2009) also observed this phenomenon in the ozonolysis of  $\alpha$ -pinene and  $\beta$ -pinene in aqueous phase.
- 130 Thus the effect of the particle acidity on the results might also be limited here, and the accurate estimates of this issue still needed further research.

Q3. It has been reported that water accelerates the decomposition of alphahydroxyalkyl-hydroperoxides (formed by SCIs + water) and alpha-acyloxyhydroperoxides (formed by SCIs + carboxylic acids) [see Zhao et al. J. Phys. Chem. A 2018, 122, 5190, Qiu et al. Environ. Sci. Technol. 2020, 54, 3890–3899]. Could this

humidity-assisted decomposition of ROOH explain the observed RH effects on SOA yield? The authors should comment on the issue in the text.

A3: Thanks for your suggestion. Water could accelerate the decomposition processes of  $\alpha$ -acyloxyalkyl-hydroperoxides and  $\alpha$ -hydroxyalkyl-hydroperoxides in aqueous phase, resulting in the formation of H2O2. The reactions of H2O2 in particles have been thought to be important and may impact the aerosols formation. A discussion about this

C3: Lines 368–377 in Sect. 3.3.3:

issue was provided in the revised manuscript.

[revised manuscript text omitted]

**Response to Reviewer #2**

We gratefully thank you for your constructive comments and through review. Below are our point-by-point responses to your comments.

**(Q=Question, A= Answer, C=Change in the revised manuscript)**

- 235 General Comments: This manuscript by Gong and Chen describes a series of laboratory experiments, aiming to elucidate the contribution of stabilized Criegee intermediates (SCIs) to the formation of SOA from limonene. The authors used a creative flow tube setup to investigate SCIs arising from ozonation of the endo- and exo-double bonds (DBs) separately. By employing an SCI scavenger in the system, the authors claimed
- 240 that they have quantified the contribution of SCI chemistry towards the SOA yields as a function of RH. The major conclusion is that water plays a complex role in the reaction system, suppressing SOA formation under low RH, while facilitating SOA formation at high RH. Over the past few years, the importance of SCI chemistry in the atmosphere has become evident. With SCI being a reactive intermediate that is difficult
- to detect, quantitative evaluations for the importance of SCI is lacking. This manuscript aims to provide quantitative information that fills our gap in understanding. The topic is timely and is within the scope of ACP. The writing and data analyses were conducted with caution. However, I have concerns regarding a few approaches and assumptions that authors employ in the study. I recommend a major revision before publication on ACP.

A: We highly appreciate your comments and suggestions. The questions you mentioned are answered as follows.

**Major comments:**

- Q1. I'm afraid that the contribution of SCI to the SOA formation may be exaggerated
  in the current setup due to the presence of a high concentration of butanol (OH scavenger). The authors provided an estimate that 12.5% of SCI has reacted with butanol, using a lower-band estimate for the SCI reactivity with alcohol. It is not convincing that the effect of butanol is "not important (Line 304)". Can the author perform an experiment with an aprotic OH scavenger (e.g., hexane) to experimentally
- 260 confirm their assumption?

A1: Thanks for your suggestion. Here to evaluate whether the addition of 2-butanol made obvious impacts on the results, we conducted experiments with cyclohexane as OH scavenger, and the results were discussed as follows.

Part of SCIs might react with 2-butanol, producing α-alkoxyalkyl-hydroperoxides and
 contributing to the observed SOA, especially when using low concentrations of AA and

water. To figure out whether the SOA formation potentials of SCIs estimated here were higher than those under the situation without 2-butanol, the experiments with cyclohexane as OH scavenger were carried out. Here three representative conditions: dry conditions, 40% RH (representing low-humidity conditions) and 80% RH

- 270 (representing high-humidity conditions), were analyzed in the endo-DB ozonolysis. The abilities of 2-butanol and cyclohexane on scavenging OH radicals were similar (Chew and Atkinson, 1996), however, the use of different OH scavengers brought different impacts on the reaction system. When 2-butanol was used, higher [HO2]/[RO2] was observed, which was thought to be more similar to the atmospheric conditions,
- 275 while adding cyclohexane resulted in a lower [HO2]/[RO2] (Docherty and Ziemann, 2003; Jonsson et al., 2008). In view of this, this study chose 2-butanol as OH scavenger, while the use of cyclohexane could provide a contrast to help us better understand the mechanisms in the reaction system.

In the experiments with cyclohexane, the concentrations of limonene and O3 were about 90 ppbv and 270 ppbv, and the reaction time was 240 s. Around 400 ppmv of cyclohexane was added to scavenge OH radicals. With the addition of cyclohexane, the SOA yields were found to be lower than those with the addition of 2-butanol, suggesting that higher concentration of HO2 radicals promoted aerosols formation (Keywood et al., 2004). Through adding different concentrations of AA (24–480 ppbv), the amount of

- 285 SCIs in the reaction system was regulated and calculated as elaborated in Sect. 3.1, and the dependence of SOA mass concentration on the amount of SCIs was shown in Fig. S8. The SCIs reactions still accounted for more than 60% in SOA formation and according to the fitting results, the SOA formation potentials of SCIs under the use of cyclohexane were even a bit larger than those under the use of 2-butanol, and the
- 290 deviations were within 12%. This phenomenon was speculated to be due to the higher concentration of RO2 radicals when using cyclohexane, promoting the reactions of SCIs with RO2.

To further determine the impact of 2-butanol, we also conducted experiments with the concentration of 2-butanol as about 150 ppmv, which was half of the previous concentration of 2-butanol used in experiments and was estimated to be sufficient for scavenging more than 99% OH radicals. It was observed that the amount of SOA formation was not impacted by the concentration of 2-butanol, and the SOA formation potentials of SCIs under a lower concentration of 2-butanol were similar with those under a higher concentration of 2-butanol. Based on the results elaborated above, we confirmed that the effect of 2-butanol on the results was limited. We have added the

Q2. The fraction of SCI reacting with water (SCII) was estimated solely from the

details on the experiments with cyclohexane in the Supplement.

10

formation of H2O2.

310

- How did the authors measure H2O2? Was it the gas phase H2O2 or particle phase?

305 - Although I agree that  $H_2O_2$  is the major decomposition product of  $\alpha$ -hydroxyhydroperoxides (product of SCI + H2O), the decomposition of  $\alpha$ -hydroxyhydroperoxides is an equilibrium process and may not always proceed completely.

-  $\alpha$ -hydroxyhydroperoxides are not the only source of H2O2. It's known that H2O2 is generated in SOA extracts, likely due to the decomposition of larger organic peroxides.

- In particular, Zhao et al. 2018, J. Phys. Chem. A reported  $H_2O_2$  arising from the decomposition of hydroperoxyester, which is formed from SCI + organic acids. Although the mechanism is not completely clear, this implies that the product of SCI + AA may also give rise to  $H_2O_2$ .

315 Citation: Ran Zhao, Christopher M. Kenseth, Yuanlong Huang, Nathan F. Dalleska, Xiaobi M. Kuang, Jierou Chen, Suzanne E. Paulson, and John H. Seinfeld, The Journal of Physical Chemistry A 2018 122 (23), 5190-5201 DOI: 10.1021/acs.jpca.8b02195

A2: Thanks for your suggestions. When we estimated the yield of  $SCI_I$  in limonene ozonolysis, the formation of gas-phase  $H_2O_2$  was measured, which was elaborated in

- 320 our previous study (Gong et al., 2018). For the detection of gas-phase H2O2, the gas samples passing through the PTFE filter were collected in a glass coil collector at a temperature of 4 °C with H3PO4 solution (pH 3.5) serving as the rinsing solution. After the collection, solutions containing peroxides were analyzed by HPLC (Agilent 1100, USA) coupled with post-column derivatization and fluorescence detection online.
- 325 Peroxides separated by column chromatography reacted with *p*-hydroxyphenylacetic acid (POPHA) to form POPHA dimers under the catalysis of hemin, and then the dimers were quantified using a fluorescence detector. With the increase of RH, it was observed that the yield of  $H_2O_2$  increased significantly from dry conditions to 70% RH, and the  $H_2O_2$  yield approached the limiting value above 70% RH, suggesting that reaction with
- 330 water suppressed other reactions of SCII. In the exo-DB oxidation, the formation of hydroxymethyl hydroperoxide was also taken into consideration. Through the box model simulation, the contribution of HO2 self-reaction to H2O2 formation was estimated to be limited. As for the reaction of SCIs with water, the products  $\alpha$ hydroxyalkyl hydroperoxides were reported to be preferential to decompose and
- 335 generate  $H_2O_2$  (Chen et al., 2016; Kumar et al., 2014). Although theoretical calculations indicated that the decomposition of  $\alpha$ -hydroxyalkyl hydroperoxides was slow, some studies proved that water and acids could significantly catalyze the process (Anglada et al., 2002, 2011; Aplincourt and Anglada, 2003; Crehuet et al., 2001), and  $H_2O_2$

formation occurred rapidly (Chen et al., 2016; Winterhalter et al., 2000). In addition,
 few α-hydroxyalkyl hydroperoxides larger than hydroxymethyl hydroperoxide were identified in gas phase, and the decomposition of α-hydroxyalkyl hydroperoxides was speculated to be totally completed.

The generation of  $H_2O_2$  from aerosols in aqueous phase, which is mainly due to the decomposition of some compounds, has received attentions in recent years (Wang et al.,

- 345 2011; Zhao et al., 2018). Here only the gas-phase  $H_2O_2$  was detected to estimate the yield of SCII, while it was still needed to analyze whether the formation of  $H_2O_2$  in SOA could impact the results. Zhao et al. (2018) reported the aqueous decomposition rate coefficients of  $\alpha$ -acyloxyalkyl hydroperoxides, whose lifetime was estimated to be about 24 min in liquid aerosols. As the reaction time in flow tube reactors was around
- 350 4 min, the  $H_2O_2$  formation from aerosols was not considered to contribute much to gasphase  $H_2O_2$  in this study. In addition, it was found that the dependence of  $H_2O_2$  yield on RH could be well simulated with the gas-phase mechanisms, confirming that the particle-phase formation of  $H_2O_2$  did not make obvious impact on the results. We have added the discussion about the formation and measurement of  $H_2O_2$  in the Supplement.
- Q3. The flow tube experiments employ tens of ppb of limonene with an excess amount of  $O_3$  for reactions. While these concentrations are rather typical for flow tube experiments, I think the author should discuss the feasibility of extrapolating their flow tube results to the real environment. As the authors point out, limonene mixing ratios are at the sub-ppb level for forest and urban environments. In my opinion, SCIs will
- 360 predominantly react with water when organic concentrations are low. Thus, the SOA formation potential of SCIs they determine in the flow tube may or may not be applicable to the ambient conditions.

A3: Thanks for your suggestion and a discussion on the effect of the concentrations of reactants has been added in the Supplement. In this study, to get enough products for
analysis in a short reaction time, both of the concentrations of limonene and O3 used in experiments were higher than those in the real atmosphere, and it was needed to consider the effect of concentrations of reactants. In the atmosphere, the concentrations of organic compounds formed from limonene ozonolysis are much smaller than those in flow tube reactors, while it should be noted that limonene-derived SCIs would not

- 370 only react with the compounds formed from limonene, they could also react with other compounds in the ambient air. In this study, we determined the rate of SCIs isomerization and reaction with other products. In the atmosphere, the organic compounds that SCIs could react with are generally carboxylic acids, carbonyls, alcohols, and RO2 radicals, and the concentrations of these compounds in forest are
- about  $10^{11}$  molecule cm-3,  $10^{11}$  molecule cm-3,  $10^{11}$  molecule cm-3, and  $10^{9}$  molecule

380

405

RO2 radicals were reported to be around  $10^{-10}$  molecule cm3 s-1,  $10^{-12}$  molecule cm3 s-1,  $10^{-14}$  molecule cm3 s-1, and  $10^{-11}$  molecule cm3 s-1, respectively (Khan et al., 2018; Lin and Chao, 2017; Zhao et al., 2017). It was estimated that the sum of the rate of SCIs isomerization and reaction with organic compounds in the atmosphere was on the same order of magnitude as that in experiments, and thus the results obtained here were considered to be feasible to the ambient conditions. We declare that further studies on

cm-3, respectively. In urban area, the concentrations of carbonyls and alcohols were reported to be higher because of the anthropogenic emissions (Vereecken et al., 2012). The rate coefficients of SCIs reaction with carboxylic acids, carbonyls, alcohols, and

385 different concentrations of reactants with the coexistence of other organic compounds would make the results more concise.

**Minor and Technical Comments:**

Q4. Line 155 "that" to "than"

A4: We have revised that.

390 Q5. Line 307 the sentence: "The reactions of SCIs with the compounds formed from SCIs scavengers would not compensate the effect of the consumption of SCIs on SOA formation." is unclear. Please rephrase.

A5: Thanks for your suggestion. We have revised this sentence as follows.

C5: Lines 335–336 in Sect. 3.3.2:

395 The major role of SCIs scavengers was consuming SCIs in the reaction system, and the effect of products formed from SCIs scavengers on SCIs reactions was not expected to be important.

Q6. I'm supportive of the authors' idea to provide an atmospheric implication of their findings by simulating three scenarios: forest, urban, and indoor. Instead of
investigating all the RH for each scenario, I wonder if authors can constrain the RH to ranges that are more relevant to each scenario? For instance, the most comfortable RH range for human occupancy in an indoor environment is between 30 to 60%. It is unlikely we see an indoor that are extremely dry or wet.

A6: Thanks for your suggestion. The relevant RH under each scenario was considered and provided in the revised manuscript.

C6: Lines 413-418 in Sect. 3.4:

In forest, the typical RH is higher than 60%, and for urban and indoor area, they are usually under low-humidity conditions (Carslaw, 2017; Vereecken et al., 2012). According to the SOA formation potential of SCIs, it is estimated that the typical

410 contribution of limonene-derived SCIs to SOA formation is  $(8.21 \pm 0.15) \times 10^{-2} \ \mu g \ m^{-3} \ h^{-1}$  in forest,  $(4.26 \pm 0.46) \times 10^{-2} \ \mu g \ m^{-3} \ h^{-1}$  in urban area, and  $(2.52 \pm 0.28) \times 10^{-1} \ \mu g \ m^{-3} \ h^{-1}$  in indoor area.

Q7. Acid anhydrides can be hydrolyzed to form organic acids. By any chance, can the acid anhydrides arising from SCI + AA be hydrolyzed at higher RH, regenerating AA?

- 415 A7: Thanks for your suggestion. Some acid anhydrides were stable against water (Rong et al., 2020), and some others could be hydrolyzed to form organic acids in aqueous phase (Fritzler et al., 2014). The hydrolysis reactions of acid anhydrides were usually accelerated significantly by some catalysts (Faria et al., 2008). In the gas phase,  $\alpha$ -acyloxyalkyl hydroperoxides formed from SCIs reaction with AA could dehydrate and
- form acid anhydrides (Long et al., 2009), and the reintroduction of water on anhydrides might not be easy to happen. The hydrolysis of acid anhydrides was expected to occur after anhydrides were taken up into atmospheric liquid water (Taatjes et al., 2019). Considering that the reaction time in this study was just a few minutes, this process was not predicted to be important.
- 425 Q8. Line 391-392 "On the other hand" is repeated in the same sentence.

A8: Thanks for your suggestion. We have revised that in the manuscript.

Q9. Line 173 - k(other) - I wonder if wall loss of SCIs should be included in k(other)?

A9: Yes. Although the wall loss of SCIs was difficult to estimate and was not discussed in detail, we think the rate of wall loss of SCIs could be included in  $k_{(other)}$ .

**Figure S8.** The dependence of SOA mass concentration on the amount of SCIs at different relative humidity (RH) in the first-generation oxidation with cyclohexane as OH scavenger.

[revised manuscript text omitted]
 H2O, which is  $5 \times 10^{-16}$  cm3 molecule-1 s-1 as derived in this study; [H2O] (molecule cm-3) is the concentration of H2O;  $k_{(SCI+SO2)}$  (cm3 molecule-1 s-1) is the rate constant of SCIs reaction with SO2, which is assumed as  $1 \times 10^{-11}$  cm3 molecule-1 s-1 (Lin and Chao, 2017); [SO2] (molecule cm-3) is the concentration of SO2;  $k_{(SCI+NO2)}$  (cm3 molecule-1 s-1) is the rate constant of SCIs reaction of SO2;  $k_{(SCI+NO2)}$  (cm3 molecule-1 s-1) is the rate constant of SCIs reaction of SO2;  $k_{(SCI+NO2)}$  (cm3 molecule-1 s-1) is the rate constant of SCIs reaction of SO2;  $k_{(SCI+NO2)}$  (cm3 molecule-1 s-1) is the rate constant of SCIs reaction with NO2, which is assumed as  $1 \times 10^{-12}$  cm3 molecule-1 s-1 (Lin and Chao, 2017); [NO2] (molecule cm-3) is the concentration 
[revised manuscript text omitted]